# Mechanistic exploration of polytetrafluoroethylene thermal plasma gasification through multiscale simulation coupled with experimental validation

Chu Chu [1], Long Long Ma [2], Hyder Alawi[1], Wenchao Ma [1] ✉, YiFei Zhu [3], Junhao Sun[4], Yao Lu[5], Yixian Xue [1] & Guanyi Chen[1,6,7]

The ever-growing quantities of persistent Polytetrafluoroethylene (PTFE) wastes, along with consequential ecological and human health concerns, stimulate the need for alternative PTFE disposal method. The central research challenge lies in elucidating the decomposition mechanism of PTFE during high-temperature waste treatment. Here, we propose the PTFE microscopic thermal decomposition pathways by integrating plasma gasification experiments with multi-scale simulations strategies. Molecular dynamic simulations reveal a pyrolysis–oxidation & chain-shortening–deep defluorination (POCD) degradation pathway in an oxygen atmosphere, and an F abstraction–hydrolysis–deep defluorination (FHD) pathway in a steam atmosphere. Density functional theory computations demonstrate the vital roles of $^1O_2$ and $\cdot$H radicals in the scission of PTFE carbon skeleton, validating the proposed pathways. Experimental results confirm the simulation results and show that up to 80.12% of gaseous fluorine can be recovered through plasma gasification within 5 min, under the optimized operating conditions determined through response surface methodology.

Polytetrafluoroethylene (PTFE), one of the most chemically and thermally stable polymers, has wide application prospects especially in chemical processes, aeronautics industries, fuel cell membranes, etc., with a global production of 309,000 tons in 2021[1–4]. However, the strong C–F bonds and stable helical conformation of PTFE make it resistant to end-of-life degradation, which can lead to the microplastic pollution[1,5–7]. Moreover, PTFE, as a kind of per- and polyfluoroalkyl substances (PFAS), is likely to be a source of low-molecular-weight PFAS (e.g., processing aids, synthesis byproducts and oligomers) due to the unsuitable disposal methods, which threatens the environment

as well as human health[8–11]. Hence, the appropriate disposal of PTFE is vital to the health of the ecological environment and by extension, to humanity.

The current disposal technologies of discarded PTFE can be roughly divided into three categories: physical separation, chemical oxidation and thermochemical conversion. Among them, the thermochemical conversion technologies, i.e., pyrolysis, gasification and combustion, outperform the former strategies two in thorough destruction and mineralization of the massive organic fluorine[9,12–14], which provides an efficient, scalable and widely available way to dispose of PTFE,

[1]School of Environmental Science and Engineering, Tianjin University/Tianjin Key Lab of Biomass/Wastes Utilization, Tianjin 300072, China. [2]School of Energy &Environment, Key Lab Energy Thermal Conversion & Control, Southeast University, Nanjing 210096, China. [3]School of Electrical Engineering, Xi'an Jiaotong University, Xi'an 710049, China. [4]Postdoctoral Programme, Guosen Securities, Shenzhen 518001, China. [5]School of Chemical Engineering and Technology, Hebei University of Technology, Tianjin 300401, China. [6]School of Ecology and Environment, Tibet University, Lhasa 850012 Tibet, China. [7]School of Mechanical Engineering, Tianjin University of Commerce, Tianjin 300314, China. ✉e-mail: mawc916@tju.edu.cn

without producing PFAS-containing wastes[9,12]. It has been shown that the optimized PTFE incineration could mineralize at least 56% of fluorine in PTFE. Furthermore, the emissions of well-known PFAS (such as PFOA) from this process (>870 °C, >4 s residence time, 0.3% PTFE by weight) are limited[10,15]. However, due to the stability of C–F bond in fluoropolymers[16], the variations in operation conditions will diminish defluorination performance, as well as reintroduce hazardous perfluorocarbons (PFCs) (such as hexafluoropropylene, perfluoroisobutylene, tetrafluoroethylene) and perfluorinated carboxylic acids ($C_3$-$C_{14}$) into the environment[9,11,13,17–21]. These byproducts can contribute to the greenhouse effect[22] and exert harmful effects on the skin, eyes, respiratory system, lung and skeleton[23–25]. Moreover, available studies fall short in quantitatively accounting for all the fluorine in the thermal treatment system, achieving a complete fluorine mass balance as well as confirming the degradation mechanism of PTFE[9,26]. The current state of research makes the researchers question the effectiveness of PTFE mineralization via thermochemical conversion technologies and the associated risks of fluorochemical byproducts. There is therefore a critical need to develop more efficient and advanced thermochemical conversion technologies for PTFE mineralization. To achieve this goal, it is essential to fully investigate the intermediate/terminal products, thermal decomposition pathway and the impact of operating conditions during the PTFE thermochemical treatment process.

Recently, there has been increasing interest in plasma gasification technology, which normally has higher thermal and electrical performances for high requirement cases[27,28]. The high temperature of electrons (in the order of 1000 K) and high concentrations of free radicals (such as •OH, •O, $^1O_2$) generated from plasma could increase the collision of particles, provide the reactive species and accelerate the rates of kinetically unfavorable reactions (such as the cleavage of C–F bond)[29–34]. The higher heat transfer rates and faster reactor start-up of plasma could improve the mineralization degree of PTFE reducing the potential of fluorinated organic gases emission[35–37]. Yao et al.[32] developed a $H_2$ plasma-assisted method to degrade polystyrene (PS) and detected that 90 wt.% PS could be degraded into $C_1$–$C_3$ hydrocarbons within 12 min. Saleem et al.[38] illustrated that the SPD plasma-based technology could mineralize 47% of fluorine from PFOA solutions. In addition, various electrical discharge plasma has been implemented to assist polymers pyrolysis/gasification or decontaminate the non-polymeric PFAS[32,33,38–43]. These studies indicated the great promise of plasma gasification for waste fluoropolymer treatment. However, so far, little information is available in the application of plasma gasification for waste fluoropolymer. Futhermore, due to the ultrafast dynamics and inherent intricacy of plasma gasification process, relying solely on experimental methods will not suffice to provide molecular-level insights into the fluoropolymers degradation mechanism during this process.

Previous research has demonstrated that molecular dynamics (MD) simulations and density functional theory (DFT) calculations can serve as valuable complementary tools to experiments, providing insights into the molecular mechanisms underlying PFAS removal and plasma treatment processes[44–47]. Wong et al. utilized advanced MD techniques, the first ab initio molecular dynamics (AIMD) simulations, to investigate the temperature-dependent degradation dynamics of PFOA on γ-$Al_2O_3$ surfaces[48]. According to density functional theory (DFT) calculations, Gao et al. proposed the potential energy surface for possible reactions during the degradation of C6/C6 PFPiA in a discharge plasma system[33]. By integrating multiscale simulations with experiments, this study systematically explores the microscopic mineralization mechanism and optimizes the macroscopic operational conditions during the PTFE plasma gasification process. The fluorine behavior and intermediate/product evolution were first investigated by reactive force field molecular dynamic (ReaxFF MD) method, which validated by relevant experiments. Then the hypothesis on carbon skeleton shortening and defluorination reaction pathway of PTFE was proposed based on products evolution and verified by DFT

calculations. Moreover, the operation conditions were optimized to maximize the turnover rate of PTFE towards inorganic fluorine through ReaxFF MD and response surface methodology (RSM). Experimental results have demonstrated that this combination of operation conditions could effectively enhance the defluorination efficiency of the PTFE plasma gasification process. The multi-scale simulation framework was shown in Fig. 1. This study offers an integrated perspective of plasma gasification process of PTFE, bridging the molecular level thermal decomposition behavior of PTFE with the macroscopic level operating condition regulation mechanism. The established protocol is generally adaptable for precise control of fluoropolymer thermochemical conversion processes and optimal design of engineering strategy for fluoropolymer wastes.

## Results and discussion
### Experimental validation of ReaxFF MD model
In order to evaluate the feasibility of the simulation model, the controlled experiments of thermal PTFE plasma degradation were carried out on a 30 kW DC thermal plasma gasifier. Fig. 2 compared the defluorination efficiency and gaseous products composition between experiment values and ReaxFF MD simulation values. Due to the low mass fraction <0.5 wt.% and low fluorine content of liquid products (<0.1%), this section focused on fluorine distribution in gaseous and solid products identified by ion chromatography (IC). The experimental fluorine distribution in the product streams was presented in Supplementary Table 1. The gaseous products composition and PFAS content in the methanol absorption solution of gaseous products were also investigated by gas chromatography-mass spectrometry (GC-MS) (Supplementary Table 2) and liquid chromatography-mass spectrometry/ mass spectrometry (LC/MS/MS, respectively (Supplementary Table 3).

As described in Supplementary Table 1, a 75%–81% fluorine mass balance closure was achieved in the experiment results. The missing F may be related to the reactions between fluorides and silicates, which are one of the main components of the refractory of gasifier. Low recovery rates of fluorine were also partly attributed to the emission of low molecular weight volatile organic fluorides from reaction system, and the loss of trace fluorides during the measurement process. As illustrated in Fig. 2, the mass proportion of inorganic fluorine in gaseous products from mixed atmosphere plasma gasification (73.72–77.35 wt.%) (Fig. 2A) was significantly higher than that of pure $O_2$ (51.26–59.84 wt.%) (Fig. 2B) or steam atmosphere (66.87–75.09 wt.%) (Fig. 2C). Furthermore, in both mixed (Fig. 2A) and steam (Fig. 2B) atmospheres, the experimental trends in the inorganic fluorine ratio closely matched the simulated values, exhibiting a high Pearson correlation coefficient (PCC) value. These results indicated a high degree of concordance between the experimental and simulated inorganic fluorine ratios in these two atmospheres. Conversely, the inorganic fluorine ratios obtained from experiments and the MD simulations under the oxygen atmosphere exhibited notable disparities, resulting in lower PCC values (Fig. 2B). These disparities can be attributed to the high proportion of $COF_2$ and ·$CFO_2$ in the products (Fig. 2C). These compounds can react with the water in the sampling absorbent liquid (·$CFO_2$ → ·F + $CO_2$; $H_2O$ + $COF_2$ → $CO_2$ + 2HF), resulting in the formation of $CO_2$ and HF, as well as an increase in inorganic fluorine contents in the actual experiment system[49–56]. The higher PCC (0.64) between experimental inorganic fluorine ratio and simulated mineralized fluorine ratio (including fluorinated C1 compounds and inorganic fluorine) supported this explanation (Supplementary Fig. 1).

The experimental dominant gaseous products were $CO_2$ and CO (86.99–90.20 wt.%) in the mixed atmosphere, while the simulated dominant gas component were $CO_2$ and ·$CFO_2$ (81.61–91.32 wt.%) (Fig. 2A). This difference could be explained that the ·$CFO_2$ intermediate is unstable and will react with steam to form inorganic fluorides and $CO_2$ in the actual mixed-atmosphere gasification system (·$CFO_2$ → ·F + $CO_2$)[49–54]. The PCC value exceeding 0.60 between the

## 1. Experimental Conditions

- Temperature, measured by OES
- Density of activate particles, simulated by numerical model
- Oxygen/carbon ($O_2$/C) ratio, steam/carbon (S/C) ratio

## 3. Experimental validation

- Fluorine distribution in gas, solid phase, measured by IC
- Gaseous products composition, measured by GC-MS
- Well-known PFAS content in gaseous products, measured by LC/MS/MS

**Initial conditions**

*Experiment validation*

## 2. Reactive force field molecular dynamic model (ReaxFF MD)

$$E_{system} = E_{bond} + E_{over} + E_{under} + E_{vdWaals} + E_{coulomb} + E_{val} + E_{pen} + E_{tors} + E_{conj}$$

- The evolution of products distribution at **macromolecular scale**
- The hypothesis of main reaction pathway
- The influence of $O_2$/C, S/C ratio and temperature

*Theoretical determination*

## 4. Density Functional Theory (DFT)

*Gaussian 16/Shermo 2.3.4/Multiwfn 3.8*

- Quantum Mechanical Investigation of energy barrier/structures/bond order at **atomic scale**

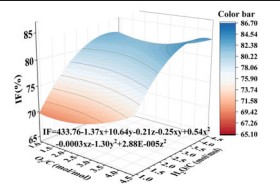

*Microscopic mechanism model at atomic and molecular scale*

**Statistical model**

## 5. Response Surface Methodology (RSM)

$$y_i = \beta_0 + \sum_{j=1}^{k} \beta_j x_{ij} + \varepsilon_i, i = 1, 2, 3, \dots n$$

- The synergic effect of $O_2$/C, S/C ratio and temperature
- The global optimization of operation conditions

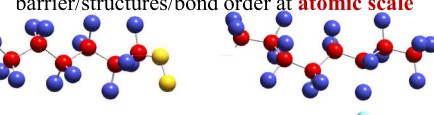

**Fig. 1 | Multiscale modeling framework for exploring the PTFE degradation performance and mechanisms by direct current (DC) thermal plasma.** (1) The experimental and initial simulation conditions, including the gas temperature and activated particle density of plasma torch, as well as the quantity of gasification agent, which were determined through optical emission spectroscopy (OES) experimental techniques. (2) The ReaxFF MD simulations for the analysis of product evolution under different atmospheric conditions, as well as the speculation on the primary reaction pathway. (3) The experimental validation of ReaxFF MD simulation result. (4) The DFT calculations for the further theoretical validation of the speculated mechanism, including the changes of bond order of key intermediates/products and the energy barriers of key reaction steps. (5) The RSM model for the investigation of synergistic influences and the optimization of operational conditions to maximize defluorination efficiency.

experimental $CO_2$ proportion and the simulated sum proportion of $CO_2$ and $\cdot CFO_2$ substantiated this hypothesis (Fig. 2A). The experimental predominant gas components were $COF_2$ and $CO_2$ (70.35−81.06 wt.%) in the oxygen atmosphere, while the simulated predominant components were $COF_2$ and $\cdot CFO_2$ (44.87 wt.%−59.90 wt.%) (Fig. 2B). This difference was also attributed to the conversion of $\cdot CFO_2$ in the actual oxygen plasma gasification process. This interpretation was supported by the observation that the PCC value of 0.67 between experimental $CO_2$ proportion and the simulated $\cdot CFO_2$ proportion, demonstrating a significant correlation (Fig. 2B). Moreover, the $CF_3O\cdot$ intermediate could be converted into $CF_4$ in the actual oxygen plasma gasification process[57,58], which ultimately led to the higher $CF_4$ content in experimental results. This explanation was substantiated by a PCC value of 0.77. As regards the steam plasma gasification process, the experimental and simulated dominant gaseous products were both $CO_2$. Furthermore, the PCC value between experimental $CO_2$ proportion and simulated value under the steam gasification reached up to 0.93, exhibiting their high consistency (Fig. 2C). However, the correlation between experimental CO (15.66−18.28 wt.%) proportion and simulated CO proportion (12.26−30.26 wt.%) were weak (PCC = 0.47). This phenomenon may be attributed to the limitations of the force field parameters utilized in this study, which might not precisely capture the behavior of C1 chemistry with sufficient precision[59]. Notably, a small amount of perfluorocyclobutane ($C_4F_8$), benzene, 1,2,3,4-tetrafluoro- ($C_6H_2F_4$) and 4-methylpent-3-enoic acid ($C_6H_{10}O_2$) were observed both in the

experimental and simulation results at the mixed and steam atmosphere (Supplementary Table 2). These results indicate that there is the potential of releasing volatile dangerous organic fluorides during the PTFE thermochemical conversion process and sufficient temperature and residence time is necessary for PTFE disposal.

Supplementary Table 3 displayed the concentrations of individual PFASs identified in the methanol absorption solution by LC/MS/MS analysis. The results showed that within procedural quantitation limits, the observed concentration of 32 PFASs in the targeted analyte list (Supplementary Table 3) were all below the detection limit (0.025 ppm), suggesting that the risk of emitting well known PFASs from PTFE thermal plasma degradation was limited, which was consistent with a previous study[10].

Overall, considering the inherent complexity of actual PTFE thermal plasma degradation process, and the likelihood of simulated intermediates undergoing further reactions to produce final experimental products in real experimental systems, we regarded that the MD simulation could qualitatively predict both the fluorine distribution in the product streams and the main gaseous product composition. It is feasible to investigate the degradation mechanism of PTFE thermal plasma gasification through ReaxFF MD simulation.

### PTFE degradation mechanism: insights from ReaxFF MD with DFT validation

The objective of this section is to present a comprehensive overview of the thermal plasma degradation process at oxygen (Fig. 3), steam

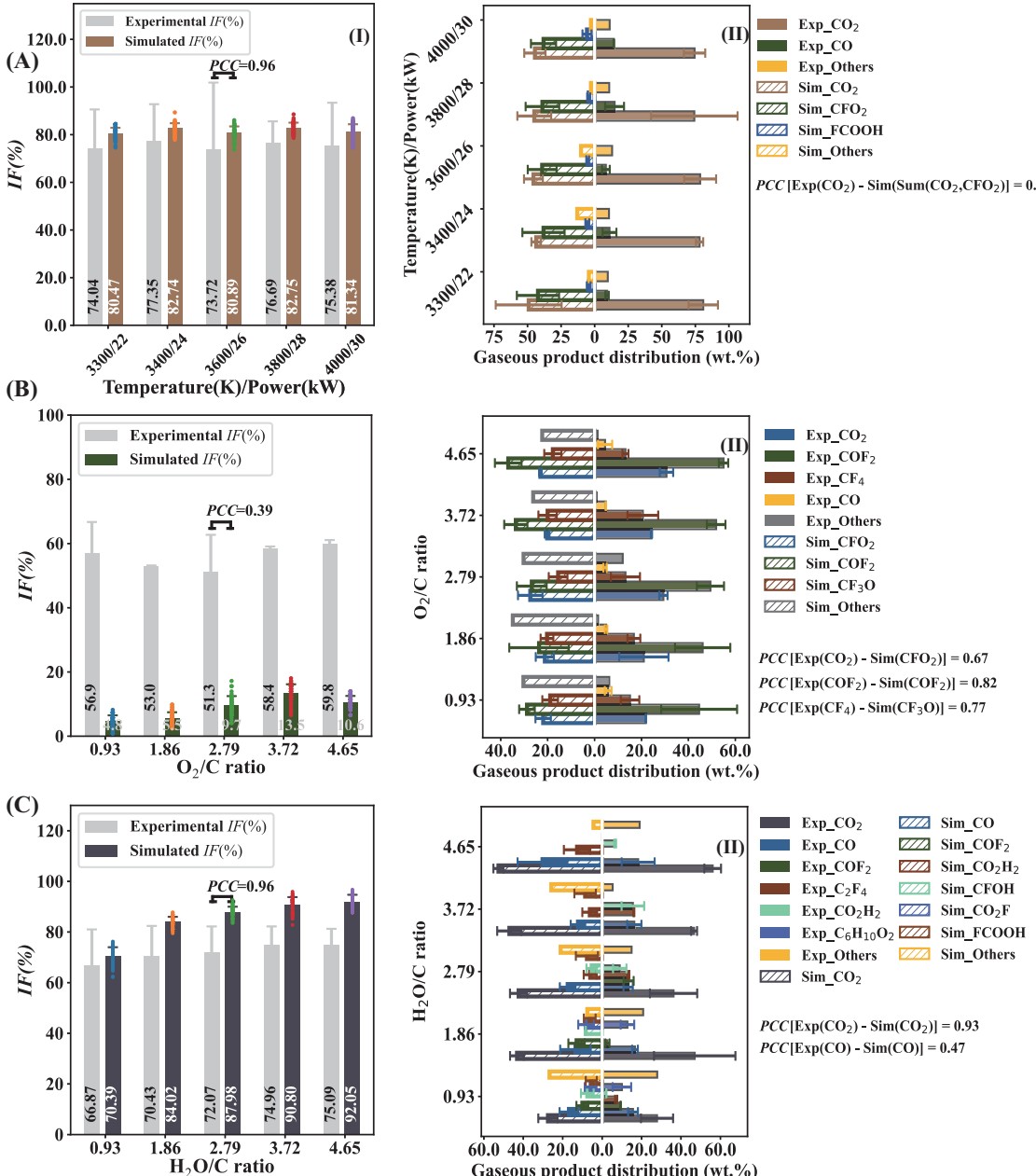

**Fig. 2 | The comparison of defluorination efficiency (I) and gaseous products composition (II) between experiment values and ReaxFF MD simulation values under different.** **A** temperature ($O_2$/C = 2.7, $H_2O$/C = 2.7), (**B**) $O_2$/C ratio (temperature = 3300 K, $H_2O$/C = 0), (**C**) $H_2O$/C ratio (temperature = 3300 K, $O_2$/C = 0). The error bars represent standard deviations (SD) derived from three measurements ($n = 3$) for experimental values, while 50 measurements for simulated values.

(Fig. 5) and mixed atmosphere. The ReaxFF MD was used to investigate the evolution of product distribution and speculate the main reaction pathway during this process at the macromolecular scale (180–1300 atoms) and experimentally relevant time scales (1000 ps). DFT was applied to offer theoretic verification for the mechanism speculations at a smaller spatial scale but at a higher level of precision. Due to the limitations of computational and storage capacities of computers, the $C_{60}F_{122}$ was selected as a representative of PTFE in ReaxFF MD simulation and $C_8F_{18}$ was chosen as the starting point for the DFT calculations in this study.

Figure 3 illustrates the product evolution, mechanism, and snapshots derived from MD simulations for PTFE oxygenic plasma gasification. It can be seen from Fig. 3A that the initial pyrolysis of PTFE chain started at 54 ps, 2829 K. As temperatures increased, the macromolecular perfluorinated radicals began to combine with $O_2$ molecules

at 58 ps, 3153 K. Then the proportion of macromolecular per-fluorinated radicals ($C_{4+}$) decreased sharply from 92.22 wt.% to 4.73 wt.% in the period between 58 ps to 75 ps. Meanwhile, the number of micromolecular PFCs ($C_{1-3}$), mainly composed of $COF_2$ and $C_2F_4$, surged from 9 to 35 in this stage. After the macromolecular decomposition stage, the temperature remained stable at around 3300 K and the micromolecular PFCs with $C_{2-3}$ were consumed and fallen to below 15 wt.% with the time increasing from 75 ps to 265 ps, while the end mineralization products such as fluorinated $C_1$ compounds, F·, $F_2$ and $CO_2$ increased gradually and accounted for over 85 wt.% of products in this stage. Ultimately, the product distribution maintained relative stability after 265 ps.

Based on the evolution of product composition and the characterization of free radical reactions, we speculated that the whole reaction process of PTFE thermal plasma degradation in the presence of

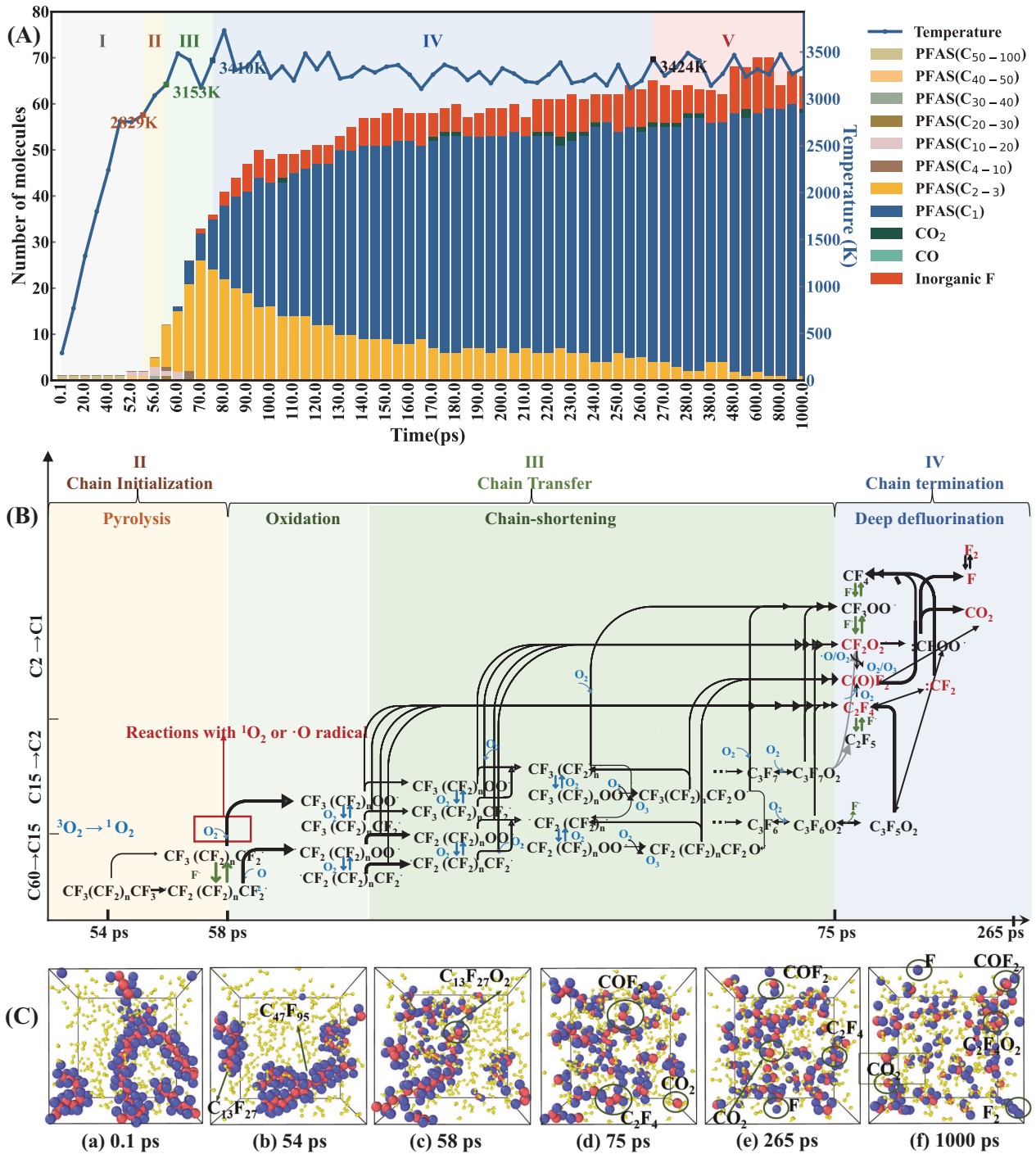

**Fig. 3 | MD simulation results of PTFE thermal plasma gasification process under oxygen atmosphere ($O_2$/C ratio = 2.79, temperature = 3300 K). A** The evolution of products distribution with treatment time. **B** The main reaction pathways in (II)−(IV) stages. **C** Snapshots of MD trajectory frames at (a) 0 ps, (b) 54 ps, (c) 58 ps, (d) 75 ps, (e) 265 ps, (f) 1000 ps. The color codes for the atoms are red: C, violet: F, yellow: O.

of oxygen could be divided into five stages as follows. The main plausible reaction pathways in II–IV stages were shown in Fig. 3B.

(I)   Heating stage: in the first 54 ps, there existed certain degrees of rotation of PTFE chain, changes in bond length and angle changes but no cleavage of bonds due to the limitations of Van Der Waals and coulomb forces.

(II)  Chain initiation stage (pyrolysis): in the period between 54 ps and 58 ps, there may occur the homolytic cleavage of intramolecular C−C bonds in PTFE and the activation of $^3O_2$, resulting in the formation of PFC ($C_{15+}$) and $^1O_2$ radicals. This hypothesis was consistent with the MD simulation results that the perfluorinated

macromolecules ($C_{15+}$) began to form and gradually increased in this stage, and furthermore, was supported by DFT calculations. According to the DFT results (Fig. 4A), the Gibbs free energy for the C−C bond dissociation in $C_8F_{18}$ molecular gradually declined from 60.43 kcal mol$^{-1}$ to −96.14 kcal mol$^{-1}$ as the temperature increase from 298.15 K to 3300 K. This result implied the rise in temperature could drive the pyrolysis reaction. After pyrolysis reaction, the Wiberg bond orders of C−C bonds in ·$C_6F_{13}$ (IM1) and $C_2F_5$·(IM2) were significantly lower than that of $C_8F_{18}$ (Fig. 4A), which verified that the pyrolysis could weaken the C−C bond and initiate the subsequent carbon backbone shortening.

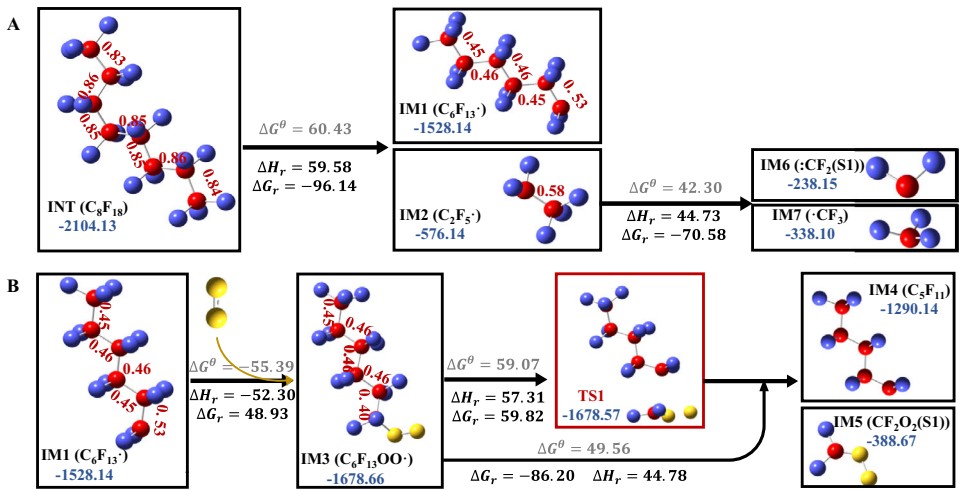

**Fig. 4 | DFT calculation results on relative energy and structure changes, energy barriers and Wiberg bond order of C−C bond (the red letters) for the two reaction processes. A** The pyrolysis of $C_8F_{18}$, **B** the oxidation of $C_6F_{13}\cdot$ and release of $\cdot CF_2O_2$ from $C_6F_{13}OO\cdot$. calculated by DFT. The color codes for the atoms are red: C, violet: F, yellow: O. represents the energy barrier in standard state (298.15 K, 1 atm). *Indicates the transition state. r represents the energy and enthalpic barrier in reaction state (3300 K, 1 atm). The energies are expressed in units of kilocalories per mole.

(III)  **Chain transfer stage (oxidation and chain-shortening):** the macromolecular PFCs gradually degraded to micromolecular PFCs ($C_{1-3}$) through two main possible ways in the period between 58 ps to 75 ps. One way was that PFC radicals react with $^1O_2$/O radicals to produce peroxyl radicals (such as $CF_3(CF_2)_nOO\cdot$), followed by the stepwise reduction of carbon chain through iterative emissions of $C_2F_4$, $CF_2O_2\cdot$ and $COF_2$. The other way was random scission of C−C bonds, similar to that of stage (II). This hypothesis of dual-path PFCs degradation pathway could explain the rapid increase of the micromolecular PFCs ($C_{1-3}$) and the existence of PFCs or perfluoroperoxyl radicals with $C_{11}$-$C_{20}$ in this stage, which was also demonstrated by DFT calculation (Fig. 4B). DFT calculations suggested that the addition of $^1O_2$ (single-state) to $C_6F_{13}\cdot$ resulted in the formation of peroxyl radicals ($C_6F_{13}OO\cdot$(double-state)) via a binding energy of 48.93 kcal mol$^{-1}$ at 3300 K (even lower at temperatures <3300 K). After that, the Wiberg bond order of the C−C bond at α location away from the OO· group decreased from 0.53 to 0.40, suggesting the higher probability for cleavage. The following DFT calculation presented that $CF_2O_2\cdot$ (singlet-state) (IM7) released from $C_6F_{13}OO\cdot$ was a barrier-less, endothermic reaction ($\triangle G_r = -86.20$ kcal mol$^{-1}$) at 3300 K, with an accessible activation energy barrier of 59.82 kcal mol$^{-1}$. These results implied that the reaction with $^1O_2$ radicals could activate the PFC intermediate and then trigger subsequent carbon-chain shortening reactions, which provided theoretical proof for the facilitating effect of $O_2$ for the defluorination of PTFE. The follow-up disassociation of $C_5F_{11}\cdot$ (IM8 → $C_3F_7\cdot$ + $C_2F_4$) (Supplementary Fig. 2), $C_3F_7\cdot$(IM9 → $CF_3\cdot$ + $C_2F_4$) (Supplementary Fig. 2), and $C_2F_5\cdot$(IM2 → $\cdot CF_3$ + $:CF_2$) (Fig. 4A) were calculated to have a very low Gibbs free energies of −140.21 kcal mol$^{-1}$, −94.92 kcal mol$^{-1}$, and −70.58 kcal mol$^{-1}$ at 3300 K, indicating the high possibility of random scission of C−C bonds of PFCs in this stage. Briefly, these DFT results supplied theoretical support for the oxidative chain scission pathway and pyrolysis chain scission pathway in this stage.

(IV)  **Chain termination stage (deep defluorination):** in the period between 75 ps to 265 ps, there were numerous intermolecular radical reactions among $^1O_2$/O radicals, peroxyl radicals, PFC radicals and alkene to produce more stable micromolecular species ($COF_2$, $C_2F_4$, F, $F_2$, $CO_2$). This speculation could explain the MD simulation results that the end mineralization products gradually predominate in the products and the mineralization rate

(MR) (calculated by Eq. (5)) value increased to 80.6 wt.% at the end of this stage. There may exist the homolytic cleavage of $C_2F_4$ to form:$CF_2$ radicals, the reaction of:$CF_2$ and $O_2$ to form $CF_2O$, and the intermolecular reaction between $CF_2O$ to form $CO_2$ and $CF_4$[60,61]. DFT calculations were conducted to assess the relative energy and structural changes for these reactions, as illustrated in Supplementary Fig. 2. The low Gibbs free energy barriers observed at 3300 K, along with supporting evidence from previous experimental research[60,61], verified the rationality of the speculation on the main pathway in this stage (Fig. 3B).

(V)  **Equilibrium stage:** after 265 ps, the degradation process was almost completed. The sum of mass proportions of fluoro- C1 compounds, F·, $F_2$ and $CO_2$ kept above 90 wt.% and the MR value rose to 95.90 wt.% at 1000 ps. The time autocorrelation functions for temperature, pressure, and total energy, presented in exponential form, were also computed for this process (Supplementary Fig. 15C). Notably, all time autocorrelation functions dropped to 0.001 within 265 ps, verifying that the reaction system reaches equilibrium within 265 ps.

The MD trajectory frames at $O_2$/C ratio of 2.79 in various stages were also consistent with the hypothesis related to the decomposition process (Fig. 3C) and suggested that the long chain PTFE molecules were degraded by plasma through pyrolysis (54 ps)−oxidation and chain-shortening (58 ps)−deep defluorination (75 ps)−equilibrium (265 ps) (POCD) pathway in the presence of oxygen. At 1000 ps, the products were more dispersed, and most are C1 compounds or inorganic fluorides.

Figure 5 presents the variation of product composition, main reaction pathway and snapshots derived from MD simulations for the PTFE steam plasma gasification process ($H_2O$/C ratio = 2.79, temperature = 3300 K). It was observed that the initial defluorination of PTFE by releasing HF started at -300 K, 0.1 ps. As the temperature increased, the cleavage of PTFE backbone started at 45 ps. With the time increasing from 45 ps to 105 ps, the number and proportion of small molecular PFCs ($C_{1-3}$) surged from 0 to 34, 0 wt.% to 77 wt.%, respectively. At the same time, the number of inorganic F also showed a rise from 2 to 31, while the proportion of macromolecular PFCs ($C_{4-51}$) decreased from 100 wt.% to 6 wt.% in this stage. After the macromolecular decomposition stage, the fluorine-free carbonaceous compounds (CO, $CO_2$, $C_{1-3}$ organics) were generated rapidly in the period between 105 ps to 285 ps, the number and proportion of which

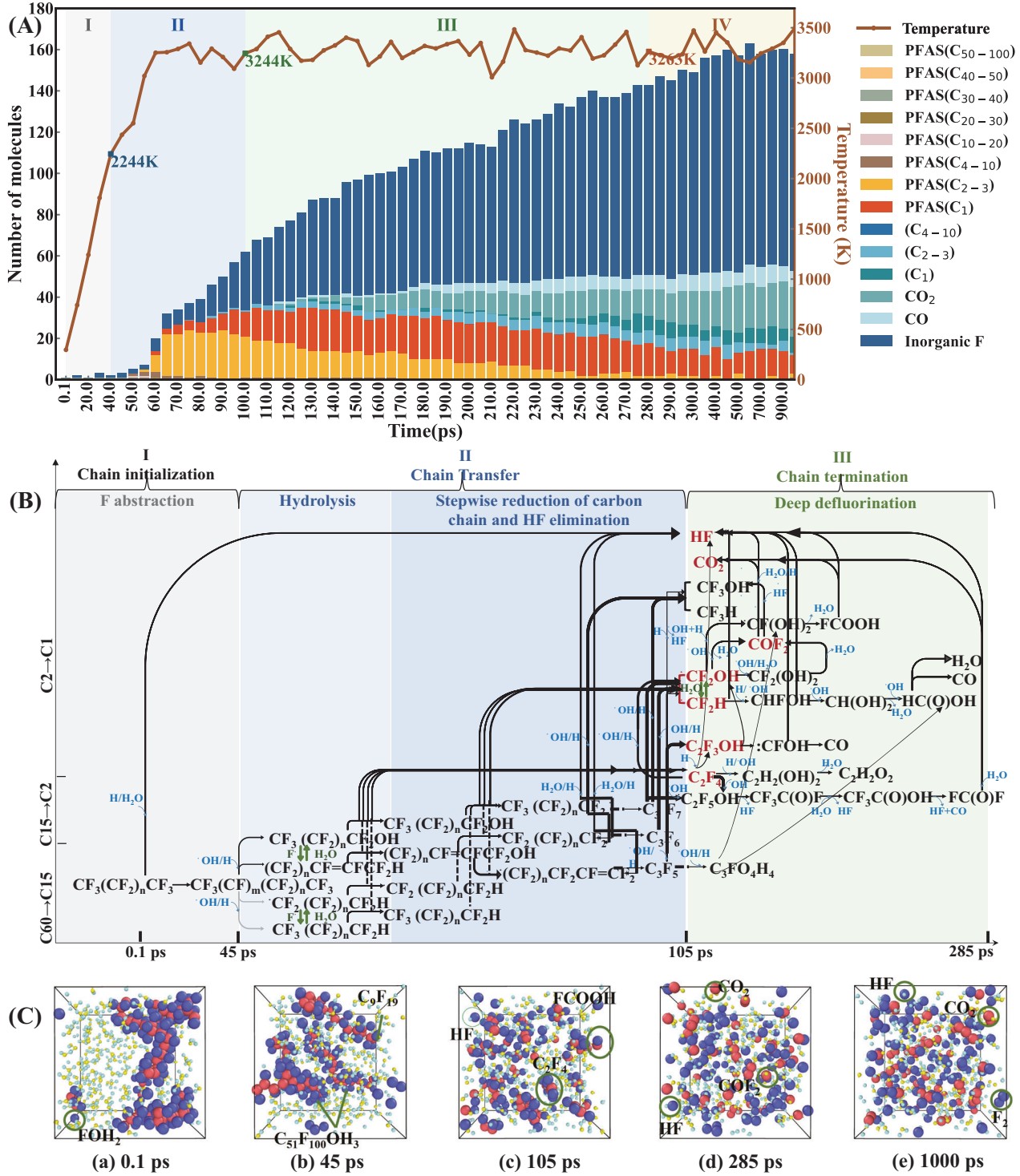

**Fig. 5 | MD results of PTFE thermal plasma gasification process under steam atmosphere ($H_2O$/C ratio = 2.79, temperature = 3300 K). A** The evolution of products distribution with treatment time. **B** The main reaction pathways in (I)–(III) stages. **C** Snapshots of MD trajectory frames at (a) 0.1 ps, (b) 45 ps, (c) 105 ps, (d) 270 ps, (e) 1000 ps. The color codes for the atoms are red: C, violet: F, yellow: O, cyan: H.

increased from 2 to 30, 5 wt.% to 27 wt.%, respectively. Meanwhile, the number and proportion of inorganic fluorides (F, HF, $H_2F_2$, HOF) shot from 31 to 90, 17 wt.% to 41 wt.%, respectively, and conversely, the proportion of small molecular PFCs ($C_{1-3}$) showed a drop of 46 wt.% in this stage. After 285 ps, the product distribution remained relatively stable.

According to the variations of product composition with time at the $H_2O$/C ratio of 2.79 (Fig. 5A), we supposed that the whole reaction

process of PTFE steam plasma gasification involved four stages as follows. The main reaction path in stages I–IV were shown in Fig. 5B.

(I) Chain initiation stage (F abstraction): in the first 40 ps, the generated H· by plasma or $H_2O$ molecules firstly attacked the fluorine atom adjacent to carbon backbone, initiated the subsequent defluorination of PTFE by releasing HF and formed PFC radicals. This speculation was in agreement with the formation of HF and perfluorinated macromolecules ($C_{15+}$) in this stage. DFT

calculations indicated that the initial F abstraction resulting HF and IM1 needed to climb a facile barrier of 23.87 kcal mol$^{-1}$ in a standard state (298.15 K, 1 atm) (Supplementary Fig. 3), thus providing the support for the formation of HF and PFAS with $C_{50+}$ in the first 0.1 ps. After F abstraction, the Wiberg bond orders of C–C bonds in $C_8F_{17}$· (IM1) were significantly lower than that of $C_8F_{18}$ (INT), demonstrating the weakened strength of C–C bonds and the higher likelihood of chain scission of PTFE. The further F abstraction from $C_8F_{17}$· (IM1) required a higher activation energy of 49.64 kcal mol$^{-1}$ in a standard state, which was also accessible for a plasma reaction system (Supplementary Fig. 3). These DFT results validated that H· radicals could promote the F abstraction reaction at low temperature and then trigger the follow-up PTFE degradation.

(II) Chain transfer stage (hydrolysis): in the period between 40 ps and 105 ps, the degradation of PFCs and the release of HF were the dominant reactions in this stage. There may be two main plausible pathways for macromolecular PFCs degradation. On the one hand, the F abstraction resulted in the cleavage of long chain PFCs. On the other hand, the carbon chains of PFCs were gradually shortened by releasing ·$CF_2OH$/·$CF_2H$ and $C_2F_4$. This speculation was in accord with the MD simulation results which showed the rise in micromolecular PFCs ($C_{1-3}$) and HF in this stage, as well as the decrease of macromolecular PFCs (Fig. 5A). This speculation was also proved by DFT calculations, shown in Supplementary Fig. 3. As illustrated, after twice F abstraction, the fission of $C_4$-$C_5$ bond in $C_8F_{16}$ (IM2) resulted in the formation of $C_4F_8$ (IM3), which required a very low free energy barrier (−14.68 kcal mol$^{-1}$ at 298.15 K and −169.54 kcal mol$^{-1}$ at 3300 K), which was much lower than direct pyrolysis of $C_8F_{18}$ molecules (60.43 kcal mol$^{-1}$ at 298.15 K and −96.14 kcal mol$^{-1}$ at 3300 K), which further confirmed the significant contributions of F abstraction to carbon chain rupture and backed up the Wiberg bond order results (Supplementary Fig. 3). DFT calculations revealed that the stepwise chain shortening of $C_4F_8$ (IM3) by releasing ·$CF_3$ (IM4) and:$CF_2$ (IM6) were barrier-less reactions with a bond dissociation free energy of −85.28 kcal mol$^{-1}$ and −52.62 kcal mol$^{-1}$, respectively (Supplementary Fig. 3), indicating that these reactions could occur easily in this stage.

(III) Chain termination stage (deep defluorination): in the period between 105 ps and 285 ps, the micromolecular PFCs (such as $C_2F_4$, $C_3F_5$, $C_3F_7OH$, ·$CF_2OH$) and free radicals (H·, OH·) reacted and collided with each other. Then substantial HF elimination, F abstraction and radical substitution reaction took place, leading to the dominant proportion of inorganic fluorides, and fluoride-free micro-molecules ($CO_2$, $C_{1-3}$) and achieving the target of defluorination. Based on DFT calculations, the Gibbs free energy and enthalpy barriers of some of the above reactions were shown in Supplementary Fig. 4, including the reactions between:$CF_2$/·$CF_3$/$C_2F_3$ and ·OH, the formation from ·$CF_2OH$ to $CF_2O$, the formation of $CO_2$ and HF and so on. The DFT results revealed that most of these reactions had low free energy barriers, demonstrating the rationality of the speculation on the main pathway in this stage (Fig. 5B). It is worth noting that while the combination of ReaxFF MD and DFT simulations in this section offers a reasonable explanation for the defluorination process of micromolecular PFCs, the conversion pathways involving fluorine-free carbon compounds (CO, $CO_2$, $C_{1-3}$) were complex and still not fully understood. Further investigation of the detailed reaction mechanisms involved in small hydrocarbons may require the use of more precise and targeted CHO force field parameters[59].

(IV) Equilibrium stage: after 285 ps, the product distribution remained relatively stable. In addtion, the MR and inorganic fluorine (IF) values (calculated by Eq.(5)) increased to 97.54 wt.% and 77.87 wt.%, respectively, at the end of this stage. The time autocorrelation

functions for temperature, pressure, and total energy, presented in exponential form, were also computed for this process (Supplementary Fig. 16C). It could be observed that all time autocorrelation functions decreased to 0.001 within 280 ps, verifying that the reaction system reaches equilibrium within 280 ps.

It can be seen from the MD trajectory at $H_2O$/C ratio of 2.79 (Fig. 5C) that in the presence of water molecules and ·H/·OH radicals, the PTFE molecules were decomposed by plasma through an F abstraction (0.1 ps) – hydrolysis (45 ps) - deep defluorination (105 ps) - equilibrium (285 ps) (FHD) pathway. At 1000 ps, most of the fluorine elements existed in the inorganic or single-carbon fluorides.

Supplementary Fig. 5 illustrates the evolution of products, the reaction mechanism, and snapshots obtained from MD simulations for the PTFE plasma gasification process in a mixed atmosphere of $O_2$ and steam ($O_2$/C ratio = 2.79, $H_2O$/C ratio = 2.79, temperature = 3300 K). It can be observed from Supplementary Fig. 5A that the whole reaction process in mixed atmosphere could also be divided into chain initialization stage (I) – F abstraction and chain cleavage (0 ps–0.1 ps), chain transfer (II) – oxidation and hydrolysis (0.1 ps–65 ps), and chain termination (III) – deep defluorination (65 ps–135 ps). These simulation results revealed that after these three reaction stages, the long chain PTFE were gradually degraded into macromolecular fluorides (C > 5), micromolecular fluorides (C1-3) and finally converted into mineralization products.

It should be pointed out that the chain initialization and initial scission of carbon skeletons in the mixed atmosphere were completed in the first 0.1 ps (Supplementary Fig. 5A), which was ~560 times and ~350 times faster than in the single oxygen atmosphere ($O_2$/C ratio = 2.79) (Fig. 3A) and steam atmosphere ($H_2O$/C ratio = 2.79) (Fig. 5B), respectively. The chain transfer stage in mixed atmosphere primarily involved the fluorine abstraction, the homolytic or hydrolytic cleavage of carbon skeletons, and a gradual shortening of the carbon chain through repeated separation of $C_2F_4$, $COF_2$, $CF_2O_2H_2$/·$CF_2O_2H$. At the end of this stage (81 ps), the fluorine in PTFE raw materials was thoroughly transformed into inorganic fluorides and micromolecular PFCs ($C_{1-3}$). Notably, according to MD simulation results, some poisonous or greenhouse gasses such as cyclo-$C_4F_8$, $C_3F_6$ and $(CF_3)_2C = CF_2$ were also generated in this stage, indicating the importance of sufficient temperature and residence time for PTFE thermal degradation. The reaction pathways of chain termination stage under the mixed atmosphere were much more complicated than that in the single atmosphere (Supplementary Fig. 5A, B). After dynamic interactions among the micromolecular fluorides and free radicals, the MR value reached 100% at the end of chain termination stage (135 ps). The product distribution stayed roughly the same in the follow-up equilibrium stage (135–1000 ps) (Supplementary Fig. 5A). The time autocorrelation functions for temperature, pressure, and total energy of this process dropped to 0.001 within 135 ps, verifying that the reaction system reaches equilibrium within 135 ps (Supplementary Fig. 17C). The higher mineralization extent, less time to reach equilibrium and more stable end products implied the promoting effect of mixed reaction atmosphere on degradation performance for PTFE.

In short, we have performed ReaxFF MD and DFT calculations to explore the molecular details of product evolution and fluorine distribution to gain a deeper understanding on PTFE plasma gasification mechanism under oxygen, steam and mixed atmosphere. Based on the product evolution results simulated by ReaxFF MD, the POCD pathway in the oxygen atmosphere and FHD pathway in steam atmosphere were proposed, in which the key steps were verified by DFT calculations.

## The influences of operational conditions: insights from ReaxFF MD and RSM

The aim of this section is to investigate the isolated and synergistic influences of operational conditions using ReaxFF MD and RSM. The

dependency of product evolution on $O_2/C$ ratio (0.93–4.65) were shown in the Supplementary Fig. 6. The main reaction process and pathway at different $O_2/C$ ratio were similar. However, the increase of $O_2/C$ ratio can improve the formation rate of $\cdot F/F_2$, and the degradation rate of PFCs with $C_{4+}$ and $C_{2-3}$, as well as shorten the time to reach the equilibrium. During the time period from 500 ps to 1000 ps, simulation results were sampled every 10 ps to calculate and compare the average product distributions, MR, and IF values during the equilibrium stage under varying $O_2/C$ ratios (refer to Supplementary Fig. 7). It can be observed that the increase of $O_2/C$ ratio was beneficial for the improvement of MR (95.92–98.87%) and IF values (4.77–10.62%). In particular, the reaction process without O· free radicals in the initial model were also simulated to compare the conventional high temperature gasification with plasma gasification (the initial simulation conditions were shown in Method Section: ReaxFF MD simulations). Supplementary Fig. 7 showed that most of the conventional thermochemical degradation process (0.93–C, 2.79–C, 3.72–C, 4.65–C) without O· free radicals in the initial simulation model had a slightly lower MR value than that of the plasma degradation process at the same $O_2/C$ ratio. This phenomenon may stem from the possible activation function of O· for $^3O_2$ molecule or PFC fragments.

As shown in the Supplementary Fig. 8, the evolution of product composition at different $H_2O/C$ ratio had the same tendency, indicating the similar reaction process and pathway at different $H_2O/C$ ratio. The rise in $H_2O/C$ ratio could enhance the release rate of HF and reduce the time to reach equilibrium stage. The average product distributions, MR, and IF values during the equilibrium stage under varying $H_2O/C$ ratio ratios are shown in Supplementary Fig. 9. At the same $H_2O/C$ ratio, most of MR and IF values from thermal plasma gasification process were slightly higher than the conventional gasification process without H· and OH· in the initial models. However, the IF value of from plasma gasification process were slightly lower than that of conventional gasification at the $H_2O/C$ ratio of 4.65, which was partly attributed to the recombination of F· and $C_{2-3}$ PFCs to form single-carbon fluorides. As the $H_2O/C$ ratio increased from 0.93 to 4.65, the MR values showed a general upward trend and reached 98.94% at the $H_2O/C$ ratio of 4.65, and the IF values increased to the maximum (92.05%) at the $H_2O/C$ ratio of 4.65.

The influences of temperature on product characteristics was discussed in the range of 3300 K–4000 K (at 1000 ps, $O_2/C = 2.79$, $H_2O/C = 2.79$). The average product distributions, MR, and IF values during the equilibrium stage under varying temperature were shown in Fig. 6A. The MR value was flat at 99.9% in the temperature range of 3300 K–4000 K. The IF value fluctuated in a range of 81%–83% with the temperature increasing from 3300 K to 4000 K. Although temperature had minor effect on the final product composition, the detailed variation trend of product distribution at different temperatures (Supplementary Figs. 5A, 10) indicated that elevated temperature could significantly accelerate the rate of HF and $CO_2$ formation in the chain termination stages (III) and shorten the time to reach equilibrium.

Based on the aforementioned single-factor analysis of the $O_2/C$, $H_2O/C$, and temperature in the PTFE degradation process, the main conclusions are as follows. The MR values from the PTFE plasma gasification process with activated radicals (O·, ·OH, ·H) were higher than the conventional gasification process. The rise in $O_2/C$ and $H_2O/C$ ratio was favorable for the MR and IF values, indicating the enhancement of defluorination performance. The dependency of MR and IF on temperature were less than $O_2/C$ and $H_2O/C$. These MD simulation results were in keeping with the experimental results (refer to Experimental validation of ReaxFF MD model Section).

To further explore the synergistic influences of these three conditions and maximize the IF and MR values, MD simulations were designed and regression analyzed through Box-Behnken design in the Design-Expert software. As shown in Table 2, all MR values under the mixed atmosphere were 100%. Therefore, this section focused on the

IF values. A multi regression model of IF values was established based on RSM and the result was obtained as follows:

$$IF = 433.02 - 1.39x + 10.57y - 0.21z - 0.21xy - 2.75 \times 10^{-4}xz + 0.51x^2 - 1.24y^2 + 2.88 \times 10^{-5}z^2 \tag{1}$$

Where IF represents the IF ratio (%), X, Y, Z represents the corresponding uncoded variables of $O_2/C$ ratio (mol/mol), $H_2O/C$ ratio (mol/mol) and temperature (K).

The results of the analysis of variance (ANOVA) (Table 1) verified the significance and reliability of this model. The $p$ values of RSM model and lack of fit are 0.0017 (<0.01) and 0.4119 (>0.05), respectively, exhibiting that this regression models were statistically significant and feasible for IF responses. The high $R^2$ value (0.9396) further validated the high fitting degree of this model and corresponding MD simulation values.

As shown in the regression model (Eq. (1)), the temperature had little impact on IF value in the parameter range of this study (3300 K–4000 K), which was consistent with the result form single effect. It can also be observed that the contribution of $O_2/C$ ratio was significantly lower than $H_2O/C$ ratio within the range of 0.9–4.65 (Fig. 6B). In general, the IF values were on the rise as the $H_2O/C$ ratio increase from 2.79 to 4.00 and then slid slowly down as the $H_2O/C$ increased from 4.00 to 4.65. Based on derivation calculations of the regression equation, the highest IF value (84.61%) was obtained when the $O_2/C$ ratio was 4.65, the $H_2O/C$ ratio was 4.00 and temperature was 3200 K. Three sets of repeated experiments under these optimal conditions were carried out to validate the simulation result. The average IF value at $O_2/C$ ratio of 4.65, $H_2O/C$ ratio of 4.00 and input power of 22 kW was 80.12%, which was higher than other experimental conditions (Fig. 2) and considered as the optimum operational conditions to obtain the highest defluorination performance.

## Method
### Modeling description
As shown in Fig. 1, in order to determine the initial simulation conditions, we first investigated the gas temperature and activated particle density of the plasma torch by optical emission spectroscopy (OES) and numerical modeling. Then ReaxFF MD simulation was employed to explore the evolution of product distribution at oxygen ($O_2/C$ ratio = 0.93, 1.86, 2.79, 3.72 and 4.65), steam ($H_2O/C$ ratio = 0.93, 1.86, 2.79, 3.72 and 4.65) and mixed atmosphere ($O_2/C$ ratio = 2.79, $H_2O/C$ ratio = 2.79) at the macromolecular scale (180–1300 atoms) and experimentally relevant time scales (-1000 ps). Next, ReaxFF MD simulation on product distribution were validated by experimental data. After that, the speculation on the main reaction pathway were proposed according to the species evolution simulated by ReaxFF MD and free radical reaction characterization. In order to offer further theoretical validation for the mechanism's speculation, DFT was used to calculate the changes of bond order of key intermediates/products, the energy barriers of key reaction steps, and the roles of key free radicals (•H, •OH, $^1O_2$) at a smaller spatial and temporal scale but at a higher precision level. Afterward, based on the Reaxff MD simulation data, the single-factor and synergic influences of $O_2/C$, $H_2O/C$ ratio and temperature were investigated using RSM.

### Initial simulation parameters: measurement of plasma characteristics
In order to determine the simulation condition of ReaxFF MD, the core temperature of plasma jet was measured by OES (Optosky ATP2400). The spectra optical emission spectra of plasma jet at different input power ($O_2/C = 0.93$) were shown in Fig. 7. Based on the comparison results of optical emission spectra of plasma jet with the standard spectrum, we determined that the core temperatures of the plasma jet at the input power of 22 kW, 24 kW, 26 kW, 28 kW, and 30 kW are

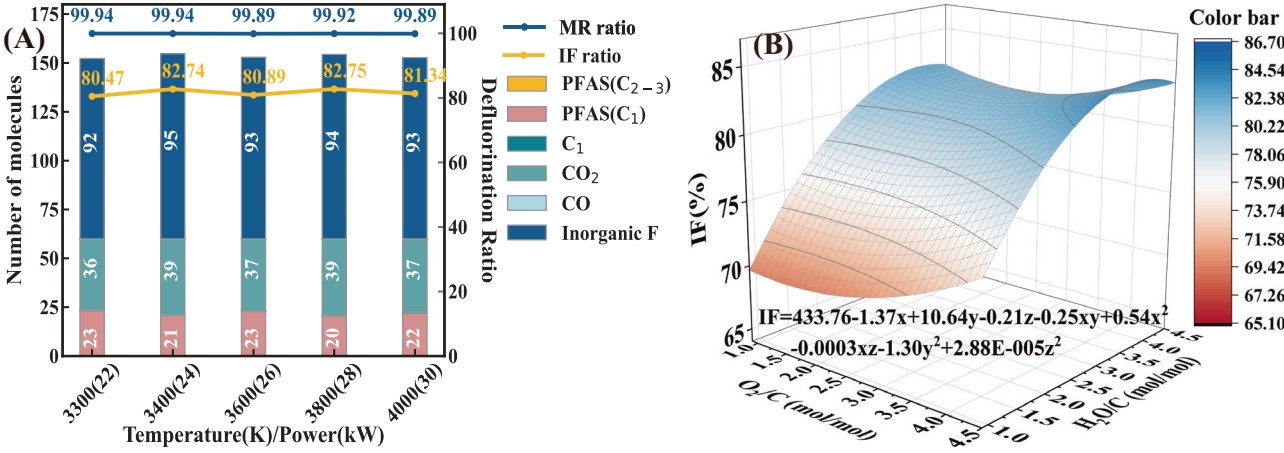

**Fig. 6 | The single and synergistic effects of operational conditions on defluorination efficiency. A** The comparison of product composition and defluorination ratio at the equilibrium stage among different temperature ($O_2$/C ratio = 2.79, $H_2O$/C ratio = 2.79). **B** 3D response surface model for representing the synergy effect of $O_2$/C ratio and $H_2O$/C ratio on IF value (temperature = 3600 K).

4500 K, 4550 K, 4650 K, 4700 K, and 4750 K, respectively. Subsequently, a two-dimensional numerical model based on the DC plasma generator structure (Supplementary Fig. 12), the core temperature, voltage, and current values of the plasma generator, was developed using Comsol software to simulate the temperature distribution of the plasma jet. As depicted in Supplementary Fig. 13, the gas temperature decreases rapidly as the distance from the center increases from 2 mm to 6 mm and remains relatively stable within the range of 6–10 mm. Given that the plasma jet has a diameter of 10 mm and the majority of the reaction zones occur at the periphery of plasma jet, we have determined the temperatures in the plasma reaction zone to be approximately 3300 K, 3400 K, 3600 K, 3800 K, and 4000 K, corresponding to input powers of 22 kW, 24 kW, 26 kW, 28 kW, and 30 kW, respectively. Hence, the final temperature parameters for the reaction systems in the MD simulation were set as follows: 3300 K, 3400 K, 3600 K, 3800 K, and 4000 K, corresponding to input powers of 22 kW, 24 kW, 26 kW, 28 kW, and 30 kW, respectively. Finally, the activated particle density of plasma jet was investigated by a zero-dimensional numerical modeling. The result of activated particle(s) density of plasma jet was shown in Supplementary Fig. 14. More detailed measurements methods as well as simulation results on temperature and activated particle density of plasma jet were shown in Supplementary Method 1, Supplementary Tables 4–9 and Supplementary Figs. 11–14.

### ReaxFF MD simulations

ReaxFF MD, developed by Van Duin et al.[62], is a force field form based on bond order. ReaxFF MD could combine the efficiency of empirical force field and the accuracy of quantum calculation, which is more suitable for simulating the complex chemical reactions in large-scale system than DFT due to the lower computational costs[63]. Reaxff MD has been demonstrated as a reliable method to simulate various thermochemical reactions at the molecular level, such as pyrolysis[64,65], gasification[66,67] and combustion[65,68]. The potential function expression of ReaxFF force field is as follows[69]:

$$
\begin{aligned}
E_{system} = &\ E_{bond} + E_{over} + E_{under} + E_{vdWaals} + E_{coulomd} + E_{val} + E_{pen} \\
&+ E_{tors} + E_{conj}
\end{aligned}
\tag{2}
$$

Where, $E_{bond}$ represents the bond energy determined by the bond order, $E_{over}$ and $E_{under}$ represent over-coordination and under coordinated energy respectively, $E_{vdWaals}$ represents the Van Der Waals interaction, $E_{coulomd}$ represents the coulomb energy between two atoms; $E_{val}$, $E_{pen}$, $E_{tors}$, $E_{conj}$ represent valence angle energy, penalty

energy, dihedral angular torsion energy and conjugation effect energy, respectively.

The molecular models were constructed using the Visualizer module in the Material Studio (MS) software (v20.1.0.2728). The thermal plasma degradation processes in different reaction atmospheres ($O_2$, steam and mixed) were simulated using LAMMPS simulation software (22Aug18) coding in conjunction with the ReaxFF method. The ReaxFF force field parameter used in this study was developed by Wood et al. for C/H/O/S/F/Cl/N systems[70].

Although the realistic PTFE polymers consist of thousands of atoms, there are limitations of computational and storage capacities of computers. Herein, the PTFE chain used in MD simulation was established from the basic molecular unit of TFE with a polymerization degree of 30. At the temperature of 298 K and pressure of $1.01 \times 10^5$ Pa, the optimized PTFE chain ($C_{60}F_{122}$), $H_2O$/$O_2$ molecules as well as free radicals (·OH, ·O, ·H) were assembled together in a cubic periodic unit by Amorphous Cell module to simulate the structure of real plasma reaction system. Notably, the ratio of free radicals to PTFE were consistent with the simulation results of the numerical model (Supplementary Method 1, Supplementary Equation 10). Supplementary Table 10 lists the detailed construction information of different reaction systems with different $O_2$/C ratio (0.93, 1.86, 2.79, 3.72 and 4.65), $H_2O$/C ratio (0.93, 1.86, 2.79, 3.72 and 4.65) and temperature (3300 K, 3400 K, 3600 K, 3800 K, 4000 K). The reaction models without O·/OH· free radicals were also constructed in the same way to compare the conventional high temperature gasification and plasma gasification. The geometric optimization of single PTFE/$H_2O$/$O_2$ molecules and periodic units were carried out through Dmol3 module in MS with the basis set of DNP and the energy convergence range of $1 \times 10^{-5}$ Hartree.

After geometric optimization, the thermal plasma degradation processes in different reaction systems were simulated using LAMMPS simulation software. The detailed simulation process involved the following steps: (1) energy minimization using conjugate gradient methods; (2) initialization of atomic velocity in order to achieve the system initial temperature to 300 K; (3) relaxation at 300 K for 1 ps under NVT ensemble; (4) heating the reaction system from 300 K to the targeted simulation temperature (3300 K–3800 K) at the rate of 50 K/ps; (3) Heat preservation for times up to 550 ps. Three dimensional periodic boundary conditions and a global time step of 0.1 fs were used in the whole simulation process. The NVT (constant volume/constant temperature dynamics) ensemble was selected and Nose-Hoover temperature coupling method was applied as the temperature control method.

**Table 1 | The ANOVA evaluation of the RSM model**

| Source | Sum of Squares | df | Mean Square | F Value | p value (Prob > F) | |
|---|---|---|---|---|---|---|
| Model | 399.88 | 9 | 44.43 | 12.22 | 0.0017 | significant |
| x-$O_2$/C | 1.37 | 1 | 1.37 | 0.38 | 0.5583 | |
| y- $H_2O$/C | 247.64 | 1 | 247.64 | 68.13 | <0.0001 | |
| z-Temperature | 0.12 | 1 | 0.12 | 0.032 | 0.8635 | |
| xy | 2.09 | 1 | 2.09 | 0.57 | 0.4732 | |
| xz | 0.17 | 1 | 0.17 | 0.046 | 0.8359 | |
| yz | 8.10E-03 | 1 | 8.10E-03 | 2.23E-03 | 0.9637 | |
| $x^2$ | 12.92 | 1 | 12.92 | 3.55 | 0.1014 | |
| $y^2$ | 71.54 | 1 | 71.54 | 19.68 | 0.003 | |
| $z^2$ | 89.26 | 1 | 89.26 | 24.56 | 0.0016 | |
| Residual | 25.45 | 7 | 3.64 | | | |
| Lack of Fit | 12.01 | 3 | 4 | 1.19 | 0.419 | Not significant |
| Pure Error | 13.44 | 4 | 3.36 | | | |
| Std. Dev. | 1.92 | | R-Squared | 0.9496 | | |
| Mean | 78.06 | | Adj R-Squared | 0.8619 | | |
| C.V. % | 2.45 | | Pred R-Squared | 0.4894 | | |
| PRESS | 217.18 | | Adeq Precision | 10.249 | | |

To assess whether the reaction systems has attained equilibrium, we computed the exponential time-autocorrelation functions, denoted as C_x(t), for temperature, press and total energy using the following expressions:

$$C(t,\tau) = \lim_{T \to \infty} \int x(t)x(t+\tau)p(x,t)dx \qquad (3)$$

Where, t represents time, indicating the starting point of observation; τ represents time delay, indicating the time interval or delay in observation; x represents the variables including temperature, press and total energy. The calculation results of time-autocorrelation functions under oxygen, steam and mixed atmosphere were shown in Supplementary Figs. 15–17, respectively. The temperature and total energy evolution of the reaction systems over treatment time in oxygen, steam, and mixed atmospheres were depicted in Supplementary Figs. 18–20, respectively.

## DFT calculations

DFT is a quantum-mechanical atomistic simulation method to calculate a wide variety of properties of atomic system. In this study, Gaussian 16 software was utilized for DFT calculations, including geometric optimization, frequency calculations, transition state search, intrinsic reaction coordinate (IRC) analysis and flexible scanning. B3LYP/6 − 31 + G(d,p) basis set was selected for geometric optimization and frequency calculations. Moreover, B3LYP/6 − 311 G** basis set was selected for the single point energy calculation. In addition, Multiwfn 3.8 and Shermo 2.3.4 program were also used to calculate the Wiberg bond level of C−C bonds and thermodynamic quantities at different temperatures[71,72]. The Wiberg bond order was proposed by Wiberg[73] to describe bond properties using ortho-normalized atomic orbitals, which was positively associated with the bond strength. The calculation expression for Wiberg bond order was as follows[74]:

$$W_{AB} = \sum_{\mu \in A} \sum_{\gamma \in B} P^2_{\mu\gamma} \qquad (4)$$

Where summation represents over atomic orbitals μ on atom A and atomic orbitals v on atom B, $P_{\mu\gamma}$ represents the corresponding density matrix element. For non-bonded pairs of atoms in a molecule, $W_{AB}$ is very low but (never zero).

## Response surface methodology (RSM)

In order to explore the interactive effects of $O_2$/C ratio, $H_2O$/C ratio and temperature on PTFE degradation performance, the RSM was utilized. RSM combines the suitable experimental/simulation design for adequate measurements of the responses, the multiple regression analysis to model the experimental/simulation responses and the optimization of input conditions to get the highest/lowest target parameters. The MD simulations in this section were designed and regression analyzed through Box-Behnken design in the Design-Expert software. The $O_2$/C ratios (0.93, 2.79, 4.65), the $H_2O$/C ratios (0.93, 2.79, 4.65), and temperatures (3300 K, 3600 K, 4000 K) were chosen as independent variables. The inorganic fluorine conversion ratio (IF) and mineralization rate (MR) values from MD simulation results (simulation time = 1000 ps) were set as corresponding responses. The expression formulas of MR and IF were shown in Product analysis Section. Herein, 17 MD simulations of 3 factors and 3 levels were carried out to investigate the synergistic effect and optimal combination of $O_2$/C, $H_2O$/C ratios and temperatures to produce the maximum IF and MR value. The values of operating variables ($O_2$/C, $H_2O$/C ratios and temperatures) and corresponding responses (IF and MR) were summarized in Table 2. After that, the statistical variance analysis (ANOVA) was conducted to verify the feasibility and reliability of RSM model, of which the results were shown in Table 1.

## Experimental validation methodology

To verify the feasibility of ReaxFF MD simulations, a fixed bed gasifier with a 30 kW direct current (DC) thermal plasma generator was used to perform the experimental study of PTFE plasma gasification. The PTFE powder (99%, 0.2 mm), obtained from MingZhe Chemicals Co., Ltd, China, was used as the raw material in this study. The reactor used in this work was exhibited in Supplementary Fig. 21 and the detailed introduction of reactor has been described in our previous study[75]. The only change in this work was that the place of the crucible for raw materials was higher and closer to the plasma generator (about 15 cm from the plasma generator) to keep the reaction region in the range of the plasma jet.

The main experimental procedure was as follows. First, the pure $N_2$ (99.99%) was conveyed into the electrode in the flow-rate range of 2.2–2.4 $Nm^3$ $h^{-1}$. Then the input power was set to targeted values (22–30 kW) and the plasma torch was ignited. After that, 200 g PTFE was supplied into the crucible in the reactor through a screw conveyor. At the same time, oxygen or steam was inputted through the entry port

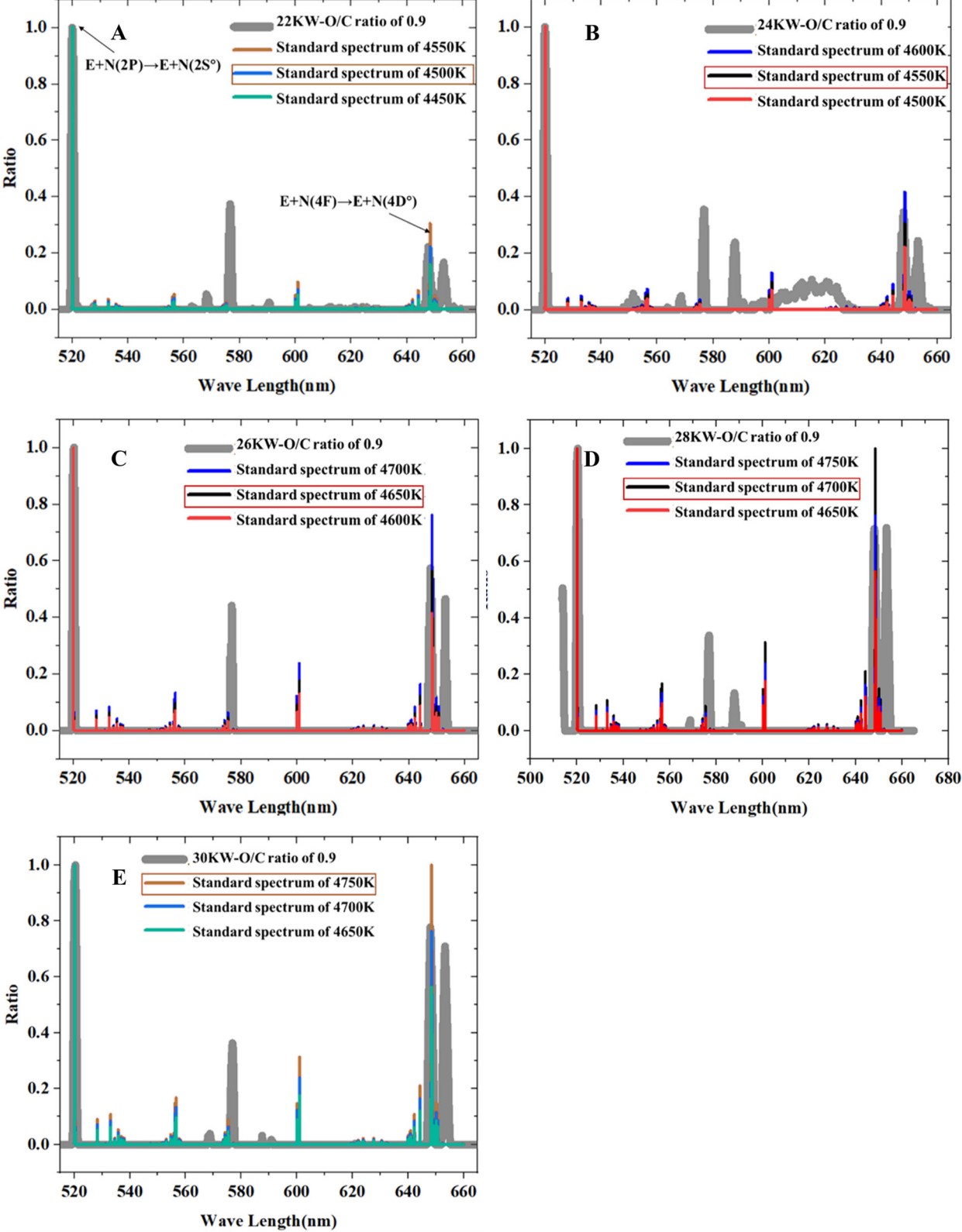

**Fig. 7 | The comparison between optical emission spectra of plasma jet and the standard spectrum under different input power (in the wavelengths range of 640–760 nm). A–E** corresponds to a specific input power: **A** Power = 22 kW, $O_2/C = 0.9$, **B** power = 24 kW, $O_2/C = 0.9$, **C** power = 26 kW, $O_2/C = 0.9$, **D** power = 28 kW, $O_2/C = 0.9$, **E** power = 30 kW, $O_2/C = 0.9$.

on the plasma reactor body in the flow-rate range of 0.5–2.5 Nm³ h⁻¹ (responding to the $O_2/C$ ratio of 0.93–4.65) and 0.22–1.08 Nm³ h⁻¹ (responding to the $H_2O/C$ ratio of 0.93–4.65), respectively. The reaction time was controlled at 5 min.

The gaseous products were collected by two different methods. The first method was based on the study done by Feng et al.[16], which was selected to gain the inorganic fluorides (HF, HOF) and PFAS (PFOA, FTOH, PFBA, etc.) through the absorption of deionized water and

**Table 2 | Box-Behnken design and the responses from MD simulation**

| Run | Factor 1 | Factor 2 | Factor 3 | Response 1 | Response 2 |
|---|---|---|---|---|---|
| | $O_2$/C ratio | $H_2O$/C | Temperature | MR | IF |
| | mol/mol | mol/mol | K | % | % |
| 1 | 0.93 | 2.79 | 4000.00 | 100 | 82.79 |
| 2 | 2.79 | 2.79 | 3600.00 | 100 | 77.87 |
| 3 | 2.79 | 2.79 | 3600.00 | 100 | 77.87 |
| 4 | 2.79 | 0.93 | 4000.00 | 100 | 72.95 |
| 5 | 2.79 | 4.5 | 4000.00 | 100 | 81.97 |
| 6 | 2.79 | 2.79 | 3600.00 | 100 | 73.77 |
| 7 | 2.79 | 2.79 | 3600.00 | 100 | 77.87 |
| 8 | 0.93 | 4.5 | 3600.00 | 100 | 82.79 |
| 9 | 2.79 | 2.79 | 3600.00 | 100 | 77.87 |
| 10 | 2.79 | 0.93 | 3200.00 | 100 | 72.95 |
| 11 | 4.65 | 0.93 | 3600.00 | 100 | 68.03 |
| 12 | 4.65 | 2.79 | 4000.00 | 100 | 83.61 |
| 13 | 0.93 | 0.93 | 3600.00 | 100 | 68.85 |
| 14 | 2.79 | 4.65 | 3200.00 | 100 | 81.97 |
| 15 | 4.65 | 4.65 | 3600.00 | 100 | 78.69 |
| 16 | 4.65 | 2.79 | 3200.00 | 100 | 84.43 |
| 17 | 0.93 | 2.79 | 3200.00 | 100 | 82.79 |

methanol. In addition, the fluorine content in deionized water and methanol were selected as the indexes of inorganic fluorine in gas products and organic fluorine in gas products, respectively. The second method adopted the 1 L TedlarTM gas sample bag to collect the $CO_2$ and small molecule fluorogases every 1 min. The collection methods of liquid and solid products were the same as our previous studies. The analysis of the weight of gas and solid, liquid products and dry gas yield had been also described in our previous study[75].

The gas products were analyzed according to the study done by Wang et al.[76] through Headspace Gas Chromatography/Mass Spectrometry (GC/MS) (Agilent 8890-5977B, USA) with a GC-GasPro column (60 m × 0.32 mm, 113–4362). The PFAS concentrations in methanol solution, liquid and solid products were determined using a Liquid Chromatography/Tandem Mass Spectrometry (LC/MS/MS) system with a Waters Acquity HSS T3 column (TSQ Quantum Ultra, Thermo Fisher, American). The extraction method of PFAS from solid products were described in detail by Feng et al.[16] and the LC/MS/MS measurement method referred to in Hao et al.[77] work. The tarted PFAS analytes were listed in Supplementary Table 11. The LC/MS spectra were shown in Supplementary Data 1. The fluorine content of deionized water was measured through ion chromatography (IC) (DIONEX, DX-120 Ion Chromatograph). The fluorine contents of methanol solution, liquid products, solid products and PTFE powder were determined by oxygen bomb digestion coupling with ion chromatography analysis. All fluorine analysis were carried out by Experimental Center of Tianjin University, China and referred to DIN EN ISO 16994:2016e12.

#### Product analysis
In order to quantitively evaluate the defluorination performance of thermal plasma degradation process, mineralization rate (MR) and inorganic fluorine (IF) conversion ratio were calculated and used as the indexes. A mixture of mineralization products including inorganic fluorides (HF, HOF, F-, $F_2$) and flurinated C1 compounds (such as $COF_2$, $CF_3OH$, FCOOH) was selected as the surrogate for final mineralization product, which was easy to be absorbed by solution and be treated by conventional air pollution control methods. A mixture of inorganic fluorides (HF, HOF, F-,) was selected as the surrogate for final defluorination product[13]. The calculation formula of MR and IF were

shown as follows:

$$MR = \frac{n_i f_i}{n_T} \times 100\% \quad (5)$$

$$IF = \frac{n_j f_j}{n_T} \times 100\% \quad (6)$$

Where, variouble $i$ indicates the species of HF, HOF, F-, fluorinated C1 compounds; $n_i$ denotes the number of $i$ molecule; $f_i$ represents the number of F atom in $i$ molecule; $n_T$ stands for the total number of fluorine in periodic unit (122); $j$ the inorganic fluorides (HF, HOF, F-), $n_j$ the number of $j$ molecules.

## Data availability
The experimental and simulated data that support the findings of this study are available in figshare Repository [https://doi.org/10.6084/m9.figshare.24312166][78].

## Code availability
The experimental and simulated data that support the findings of this study are available in Zenodo Repository [https://doi.org/10.5281/zenodo.10448083][79].

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

## Acknowledgements

This work was supported by Natural Sciences Foundation of China [52370134, 72261147460]. This work was supported by National Key R&D Program of China [2020YFC1908604].

## Author contributions

Wenchao Ma and Chu Chu directed the overall project, conceived the idea and designed the experiment. Chu Chu carried out the experiments, conducted the multiscale simulation, and wrote the draft manuscript. Long Long Ma and Hyder Alawi contributed to the design of experiments and the writing of manuscript. YiFei Zhu contributed to the characterization of plasma. Junhao Sun, Yao Lu and Yixian Xue helped with the multiscale simulation and mechanism analysis. Guanyi Chen helped with the design of experiments and the writing of manuscript. All authors read and commented on the manuscript.

## Competing interests
The authors declare no competing interests.
