## [Peer Review File · Nature Communications]

Mechanistic Exploration of Polytetrafluoroethylene Thermal Plasma Gasification through Multiscale Simulation Coupled with Experimental ValidationREVIEWER COMMENTS

Reviewer #1 (Remarks to the Author):

In this study, the author used macroscopic experiments and ReaxFF simulations, as well as DFT calculations, to validate the effectiveness of using plasma pyrolysis to decompose Polytetrafluoroethylene (PTFE), achieving a higher degradation rate than traditional treatment methods. And from a microscopic perspective, the decomposition path and mechanism of PTFE under plasma pyrolysis conditions were revealed, which is of great significance for the treatment of PTFE. Before being accepted, the author needs to consider making modifications to the following issues:

1. In line 32, page 2 and in Figure 1, the expression "S/C ratio" is confusing, it's better to express as H₂O/C ratio.
2. There is no relaxation step in description part of ReaxFF operation steps. For the ReaxFF simulation system, it is necessary to conduct sufficient relaxation before formal simulation.
3. In line 139, page 7, "2" in "O₂" requires subscripts; In line 452, page 21, "22kV" should be 22 kW.
4. In Figure 4, It is best to unify the energy unit as kCal/mol. In addition, it is necessary to present the relevant transition state structure. Is the free energy barrier of bond breaking obtained through flexible scanning? Moreover, the Gibbs free energy barrier of IM1 reaction with oxygen is -55.39kCal/mol, while the Gibbs free energy change is 48.93kCal/mol, which is impossible and needs to be checked.
5. In line 261, page 12, what evidence infers that the intermolecular reaction of COF₂ generates CO₂ and CF₄?

Reviewer #2 (Remarks to the Author):

In this submission to Nature Communications, the authors integrate multi-scale computational simulation with plasma gasification experiments to systematically investigate the microscopic thermal decomposition pathway and efficient macroscopic treatment strategy for PTFE waste. In particular, the authors use classical molecular dynamic (MD) simulations (further comments on their use of MD calculations are given below). The authors propose that PTFE could be degraded through a pyrolysis – oxidation & chain-shortening – deep defluorination (POCD) pathway in the oxygen atmosphere, and through an F abstraction – hydrolysis - deep defluorination (FHD) pathway in the steam atmosphere. The authors complement their study with static density functional theory calculations demonstrated the vital roles of IO_2 and $\cdot\text{H}$ radicals on the scission of PTFE carbon skeleton (further comments on their static DFT calculations are given below). The authors find that the experiment results validated the feasibility of the simulation.

I find this manuscript to be of interest to computational/experimental researchers in environmental contaminants as well as readers of Nature Communications. As such, I am generally supportive of publication with a few minor edits that should be incorporated. In particular, there has been review

articles and research articles on the accuracy and time scales of classical vs quantum simulations of environmental contaminants: Chem. Rev. 2021, 121, 9450–9501, Environ. Sci. Technol. 2023, 57, 6695–6702, and Environ. Sci. Technol. 2022, 56, 8167–8175. Specifically, the authors of the current paper start with classical-based ReaxFF simulations and later move to static DFT calculations. However, the review and research papers mentioned previously show that DFT-based dynamics are typically more accurate than classical MD for probing decomposition mechanisms of contaminants. In addition, static DFT calculations can often miss relevant species due to their static nature (which can be ameliorated with DFT-based dynamics). I am not necessarily asking the authors to carry out DFT-based dynamics, but the previous work using DFT-based dynamics on chemical contaminants should be noted as recent work in this area. With these minor edits, I would be willing to re-review this manuscript for Nature Communications.

Reviewer #3 (Remarks to the Author):

This manuscript describes an integrated experimental and multiscale simulation study of PTFE, an important recalcitrant yet highly popular material. This is an interesting study, and the ReaxFF reaction analysis is very well performed, but I have several issues that I believe the authors should address.

- Grammar. Overall, the grammar level is acceptable – but the authors often incorrectly use the word ‘the’, especially in their section headings. For example, ‘2.1 The experimental validation of the Reaxff model’ – in this case, ‘The’ should be removed – since ‘The’ indicates a uniqueness (like ‘The only experimental validation of the Reaxff model’). Note – also Reaxff should be spelled ReaxFF.
- I am concerned about the very high temperatures used in the ReaxFF simulations (up to 4000K). I appreciate that the authors seek to validate these temperatures by comparisons with experimental plasmas – but the experimental temperature equivalents are all deeply hidden in the SI material – the authors should at least mention these experimental temperatures in the main text. Do the authors consider these temperature realistic? At temperatures approaching 4000K electrons should be able to go to excited states – which is not described by ReaxFF.
- Why was the ReaxFF simulation performed for 550 picoseconds? Was this because by that time all the chemistry had converged in the ReaxFF simulations – or was there a different reason to end the simulation at that time? 550 picoseconds is not a very long ReaxFF simulation – especially since the simulation boxes are relatively small.
- The force field used in this study – the Wood et al ReaxFF parameter set – may have issues with the C1-chemistry, in particular CO reactions with radicals. This was described and corrected by Ashraf and co-workers (Ashraf et al. J.Phys.Chem. A, 2017, 121, 1051-1068.) Can the authors discuss how the CO-chemistry issues, as described by Ashraf and co-workers, may affect their results?

Reviewer #4 (Remarks to the Author):

The authors calculated the degradation mechanism of PTFE by thermal plasma using ReaxFF and DFT, and compared the degradation species with the results of thermal plasma experiments. The main motivation for this study is related to the establishment of a physicochemical strategy to overcome the indispensability of the PTFE. The insight brought by MD simulations showed the possibility of the existence of P-O-C-D and F-H-D pathways, and it was verified from DFT that the scission of such a carbon skeleton is caused by $\cdot\text{O}_2$ and $\cdot\text{H}$ radicals. A few minutes of plasma treatment experiments demonstrated that significant levels of fluorine were converted to inorganic compounds.

The research topic of this manuscript clearly addresses important issues of the day, and the message of the results is also clear. Therefore, it is expected to form a readership in various fields. However, this study is basically an investigation on the process of figuring out the decomposition pathway of the PTFE by thermal energy from multiple simulations and quantitatively comparing it with experimental observations. Methodologically, the multiscale model proposed in this study does not seem to have academic progress or novelty in the field of computational science. Therefore, the inventive step of the present manuscript can be found limitedly only in the industrial issue of eco-friendly decomposition of PTFE. Overall, the reviewer recommends that the authors thoroughly respond to the comments below before re-evaluation.

Major 1) The reviewer believes that the term 'global optimization' on Line 90 should be used very carefully in studies involving computational approaches. Both DFT and ReaxFF MD simulations have their own limitations as models and solvers, and we cannot guarantee that the obtained pathways and the associated energies are placed at the global minima. The author's intention in this sentence is probably to emphasize that the operation condition to maximize the effect of C-F scissoring was found from simulation. Although extensive computational results have been performed, the cases are still within a finite compositional range, so it is a leap to discuss the findings as the result of 'global optimization' of the experimental conditions.

Major 2) The authors compared the ReaxFF simulation results of the distribution of fluorocarbon decomposition species in three different environments with the experimental IC results. The reviewer basically agrees with the authors' assertion that the results in Figure 2 show that the qualitative trends of experiments and simulations are consistent. However, there is a logical leap to conclude that the validity of the simulation model itself has been verified based on these results. Above all, only one optimal condition found from the proposed RSM model has been experimentally verified for thermal decomposition of PTFE, so it cannot be said that the value is an 'experimentally optimal condition'. In other words, the RSM model established from the multiscale simulation finds one of the appropriate improvement conditions under the given conditions, not the global optimum condition of the thermal plasma process. Therefore, the abstract, introduction, and title that demonstrated that an appropriate operation condition was found from simulation conditions should all be modified.

Major 3) Continuing from the comments immediately above, the authors arbitrarily mix qualitative and quantitative comparisons when discussing the differences between the experimental and simulated results presented in Figure 2. For example, the author argued that COF_2 and $\cdot\text{CFO}_2$ appearing in

ReaxFF are the result of not considering the reaction with water. If so, does each reaction have such a self-evident chemical pathway that the experiment (or simulation considering the reaction with water) yields a reaction rate close to 100%? It is necessary to explain, at least qualitatively, what state the COF₂ and dot_CFO₂ products will remain in the real environment. Similarly, CF₃O occupies a significant proportion in the MD simulation, whereas CF₄ occupies a similar proportion in the experiment. No discussion of this can be found in the main text. A similar issue is raised with the dot_CFO₂ results in Figure 2A. Is the contribution of dot_CFO₂ in the MD simulation completely explained by the CO₂ in the experiment?

As far as the reviewer understands, the results in Figure 2 are the only part in this manuscript in which direct comparisons with experimental data have been made, and are therefore of great importance. Nevertheless, in the present manuscript, the reasons for the discrepancies between the data obtained from the simulation and the experimental values are limited to listing. The reviewer asks that the authors specify precisely in the text what part of the results observed experimentally this simulation succeeded in reproducing and what findings were captured accordingly.

Major 4) Regarding the reactive MD simulations, the reviewer concerns that the overall time integration applied to the MD models is too short. In particular, it is necessary to secure additional validity for the simulation methodology that causes thermal decomposition by raising the temperature of the model. The addition of supplementary simulation results is needed to address the following questions.

- How was the initial density of the presented model determined? In the MD recipe presented in the manuscript, there is no process to converge the density of a given PTFE system.

- Are the prepared models capable of smooth temperature control by Langevin thermostats until reaching a level of 4000 K? Present the temperature profile provided as input during the thermal decomposition simulation and the resulting total energy and the temperature change profiles of the system.

- In the opinion of reviewers, 50K/ps is too harsh and too fast a heating condition. This can cause some important events of the system's pyrolysis species to be missed or overestimated (such as the recombination of temporarily broken molecules). For a representative model, please prove with data that the considered heating rate condition is slow enough to calculate the profile of the degraded species ratio and therefore exhibits high convergence.

- As stated by the authors, the molecular weight of PTFE covered by the MD simulations is clearly too small for the experimental conditions. Additional data is needed to justify that the effect of molecular weight differences does not need to be carefully considered in this study. One solution seems to be to model an infinite length structure of self-connected PTFE chains beyond periodic boundary conditions and perform thermal decomposition simulations.

Major 5) The supplementary file contains a lot of discussion and related background on auxiliary results. However, the organization of the text is too incomplete, and each sentence is not easily read. Although all of this is supplementary material, a minimum level of readability should be ensured.

Minor 6) What does 'Operational variations' on Line 53 mean? Does that phrase refer to process conditions that deviate from the optimized incineration process mentioned in the previous sentence on Line 49? If so, why should we consider such variations? Or, is this sentence intended to emphasize that the risk of yield degradation is too high for even slight deviations from optimized conditions? The intention is not clearly understood only with the contents presented in the manuscript, so it seems that the contents need to be specified or modified.

Minor 7) The sentence placed on Lines 57 to 60 seems to be about the difficulty of quantitative detection of the thermal decomposition of PTFE. However, it does not seem to have much to do with the risk of byproduct or mineralization effectiveness. In other words, this sentence interrupts the flow of content before and after.

Minor 8) The order of text in the Methods section should follow the workflow presented in Figure 1A. From the reader's point of view, the present manuscript first encounters validation of the main data (Figure 2) and experiments without fully understanding the modeling environment and conditions of the ReaxFF simulation. Because of this, the reviewer felt that the manuscript was less readable.

Minor 9) Figures 1B-E should be drawn separately from Figure 1A. These data are related to Section 3.1 and are not suitable for presentation along with the modeling overview in the introduction.

Suggestion 10) Too many typos related to abbreviations and jargon (PFTE -> PTFE (Line 50); Molecular dynamics -> Molecular dynamics (Line 87); Reaxff or ReaxFF? (Line 106); The origin column -> the orange (?) column (Line 132). Also, it looks like the colors referring to COF2 and dot_CFO2 in the text are reversed.). Typos throughout the manuscript must be corrected before entering the next round.

Suggestion 11) The abbreviation 'SPD' on Line 76 appears only once in the entire manuscript. It is recommended not to specify abbreviations for terms that appear infrequently.

Suggestion 12) It is recommended to remove the 3D plot presented in the area introducing the RSM in Figure 1A. The labels on each axis are too small to read. Replacing them with simplified icons representing each task is also recommended.

Suggestion 13) The data and caption provided in the supplementary file are incomplete. The details to be fixed are:

- There are no yellow beads in Figure S2, but cyan beads instead, which are not described in the caption. The caption of Figure S3 also still does not define the cyan beads.
- It is believed that the final temperature on the horizontal axis of the data shown in Figures 5B and S4B must exceed 4000 K. The current data is rather lower than the previous temperature, and if it is a typo, it needs to be corrected.
- The lower part of the data plot in Figure S8 is cut off, so the horizontal axis and part of the data are not

visible.

- Please clearly indicate the information in the sub-figures from (a) to (n) of Figure S10 in the captions.

Itemized response to each review comment

Reviewer # 1

[Issue 1] In line 32, page 2 and in Figure 1, the expression “S/C ratio” is confusing, it’s better to express as H₂O/C ratio.

[Response] **Accepted.** The expression of the S/C ratio has been replaced with H₂O/C accordingly throughout the revised manuscript. As shown below:

- **Line 31, P. 2:** *Experiment results validated feasibility of the simulation and shows that up to 80.12% of fluorine could be recovered as gaseous inorganic fluorides by plasma treatment within 5 minutes, under the global optimized conditions simulated by response surface methodology ($O_2/C=4.65$, **H₂O/C** ratio of 4.00 and input power of 22 kW).*
- **Figure 1, P.5:** *The three-dimensional diagram of the optimized structure of typical periodic units for (C) O_2/C ratio of 2.79 (D) H_2O/C ratio of 2.79 and; (E) the mixed atmosphere reaction system (O_2/C ratio=2.79, **H₂O/C** ratio=2.79).*

[Issue 2] There is no relaxation step in description part of ReaxFF operation steps. For the ReaxFF simulation system, it is necessary to conduct sufficient relaxation before formal simulation.

[Response] **Accepted.** In fact, we performed a 1ps relaxation process at 300K using the NVT ensemble in each case. Sorry for not presenting the results in the previous version. The relevant description of relaxation step has been added in the Section 3.1.2.

- **Line 570, P. 27:** *The detailed simulation process involved the following steps: (1) energy minimization using conjugate gradient methods; (2) initialization of atomic velocity in order to achieve the system initial temperature to 300K; (3) relaxation at 300K for 1ps under NVT ensemble;*

[Issue 3] In line 139, page 7, “2” in “O₂” requires subscripts; In line 452, page 21, “22kV” should be 22 kW.

[Response] **Accepted.** We have double checked and corrected the similar subscript and unit errors throughout and made the following corrections.

- **Line 139, P. 7: Figure 2.** The comparison of defluorination efficiency (left) and gaseous products composition (right) between experiment values and ReaxFF MD simulation values under different (A) temperature ($O_2/C=2.7$, $H_2O/C=2.7$), (B) O_2/C ratio (temperature=3300K, $H_2O/C=0$), (C) H_2O/C ratio (temperature=3300K, $O_2/C=0$).
- **Line 459, P. 21:** The average IF value at O_2/C ratio of 4.65, H_2O/C ratio of 4.00 and input power of 22 kW is 80.12%, which is higher than other experimental conditions.

[Issue 4] In Figure 4, It is best to unify the energy unit as kCal/mol. In addition, it is necessary to present the relevant transition state structure. Is the free energy barrier of bond breaking obtained through flexible scanning? Moreover, the Gibbs free energy barrier of IM1 reaction with oxygen is -55.39kCal/mol, while the Gibbs free energy change is 48.93kCal/mol, which is impossible and needs to be checked.

[Response] Accepted. As shown in Fig. 4 and Figs. S1-S3, we have standardized the energy units used in our DFT calculations to kcal/mol. Furthermore, the XYZ coordinates of the optimized DFT structures, along with the corresponding energies for reactants, intermediates, and transition states, are provided in Data S1 (from Line 33, Page 6, to Line 881, Page 33). As shown below:

- **Figure 4.** Relative energy and structure changes, energy barriers and Wiberg bond order of C-C bond (the red letters) for the pyrolysis of C_8F_{18} , the oxidation of C_6F_{13} and release of $\cdot CF_2O_2$ from $C_6F_{13}OO\cdot$ calculated by DFT. The color codes for the atoms are red: C, violet: F, yellow: O. θ represents the energy barrier in standard state (298.15K, 1atm). *

indicates the transition state. r represents the energy and enthalpic barrier in reaction state (3300K, 1 atm). The energies are expressed in units of kilocalories per mole.

- **Data S1 XYZ Coordinates of Optimized DFT Structures and Corresponding Energies (θ represents the energy barrier in standard state (298.15K, 1atm). * indicates the transition state. r represents the energy and enthalpic barrier in reaction state (3300K, 1 atm)**

TS1

$$EE[B3LYP/6-311G^{**}] = -1677.22176 \text{ Ha}$$

$$H_{\text{corr}}^{\theta} [B3LYP/6-31+G(d,p)] = 0.09956 \text{ Ha}$$

$$G_{\text{corr}}^{\theta} [B3LYP/6-31+G(d,p)] = 0.02369 \text{ Ha}$$

$$H^{\theta} = EE + H_{\text{corr}}^{\theta} = -1677.12221 \text{ Ha}$$

$$G^{\theta} = EE + G_{\text{corr}}^{\theta} = -1677.19807 \text{ Ha}$$

$$H_{\text{corr}}^r [B3LYP/6-31+G(d,p)] = 0.63100 \text{ Ha}$$

$$G_{\text{corr}}^r [B3LYP/6-31+G(d,p)] = -1.35270 \text{ Ha}$$

$$H^r = EE + H_{\text{corr}}^r = -1676.59077 \text{ Ha}$$

$$G^r = EE + G_{\text{corr}}^r = -1678.57446 \text{ Ha}$$

Where θ represents the energy barrier in standard state (298.15K, 1atm), * indicates the transition state. r represents the energy and enthalpic barrier in reaction state (3300K, 1 atm)

C	1.54022400	0.90190000	-0.00098900
C	3.61563300	-0.86783500	0.11590400
F	1.68217100	1.65583700	-1.07070800
F	1.57400700	1.59121600	1.12695600
F	3.14104800	-2.03396800	-0.20999400
F	3.62505000	-0.62277400	1.38347200
O	4.17312600	-0.16659800	-0.74741300
O	4.65975900	1.08403700	-0.31124200
C	-1.00426000	0.47904100	0.07595000

C	0.42129700	-0.12093500	-0.07637400
F	-1.07796700	1.57813400	-0.71233500
F	-1.15742900	0.85285000	1.36810900
F	0.61497500	-1.02200200	0.92124100
F	0.51162600	-0.74937900	-1.27314300
F	-3.76924800	1.24852500	-0.18477500
C	-3.55583300	-0.02830800	0.16044600
C	-2.15365300	-0.48431200	-0.31843100
F	-3.65821200	-0.14834000	1.48780300
F	-4.48518900	-0.79693900	-0.41763000
F	-1.91030200	-1.70456700	0.21563600
F	-2.18425600	-0.58379400	-1.66460900

To compute the free energy barrier associated with bond breaking, the flexible scanning is first performed to determine whether a transition state exists. If no transition state is identified (e.g., INT \rightarrow IM1+IM2, as shown in Fig. 4), the free energy barrier for bond breaking is calculated by using the Gibbs free energy difference between the reactants and products following geometry optimization. Conversely, in cases where a transition state exists (e.g., IM4 \rightarrow IM8+PT1, as shown in Fig. S1), the free energy barrier is determined based on the Gibbs free energy difference between the transition state and the reactants.

In addition, the Gibbs free energy barrier of IM1 reaction in standard state (298.15 K, 1atm) is -55.39 kCal/mol, while it changes to 48.93 kCal/mol in reaction state (3300 K, 1 atm). This result could be attributed to the larger decrease in the free energy at 3300 K of the reactants (1.258 Ha for C₆F₁₃ and 0.286 Ha O₂) compared to that of the products (1.378 Ha for C₆F₁₃O₂).

[Issue 5] In line 261, page 12, what evidence infers that the intermolecular reaction of COF2 generates CO2 and CF4?

[Response] Accepted. We apologize for the confusion caused by our previous statement. The validation of this reaction pathway was not based on DFT calculations but on previous experimental studies. The relevant references have now been incorporated into the revised manuscript.

- **Line 4, P.13:** *There may exist the homolytic cleavage of C₂F₄ to form :CF₂ radicals, the reaction of :CF₂ and O₂ to form CF₂O, and the intermolecular reaction between CF₂O to form CO₂ and CF₄^{61,62}. DFT calculations were conducted to assess the relative energy and structural changes for these reactions, as illustrated in Fig. S2. The low Gibbs free energy*

barriers observed at 3300K, along with supporting evidence from previous experimental research^{61,62}, verified the rationality of the speculation on the main pathway in this stage (Fig.3(B)).

References

61. Hasegawa Y, Otani R, Yonezawa S, Takashima M. Reaction between carbon dioxide and elementary fluorine. *Journal of Fluorine Chemistry* 128, 17-28 (2007).
62. Pritchard GA, Amphlett JC, Dacey JR. The reaction $2\text{COF}_2 \rightarrow \text{CO}_2 + \text{CF}_4$ and the heat of formation of carbonyl fluoride. *The Journal of Physical Chemistry* 75, 3024-3026 (1971).

Reviewer # 2

[Issue 1] In particular, there has been review articles and research articles on the accuracy and time scales of classical vs quantum simulations of environmental contaminants: Chem. Rev. 2021, 121, 9450–9501, Environ. Sci. Technol. 2023, 57, 6695–6702, and Environ. Sci. Technol. 2022, 56, 8167–8175. Specifically, the authors of the current paper start with classical-based ReaxFF simulations and later move to static DFT calculations. However, the review and research papers mentioned previously show that DFT-based dynamics are typically more accurate than classical MD for probing decomposition mechanisms of contaminants. In addition, static DFT calculations can often miss relevant species due to their static nature (which can be ameliorated with DFT-based dynamics). I am not necessarily asking the authors to carry out DFT-based dynamics, but the previous work using DFT-based dynamics on chemical contaminants should be noted as recent work in this area.

[Response] Accepted. We have reviewed the publications you shared regarding DFT-based dynamics in the field, and the previous work using DFT-based dynamics on PFAS disposal has been added in the Section Introduction:

- **Line 83-93, P. 4:** *Previous research has demonstrated that molecular dynamics simulations and density functional theory calculations can serve as valuable complementary tools to experiments, providing insights into the molecular mechanisms underlying PFAS removal and plasma treatment processes^{45,46,47, 48}. Wong et al. utilized advanced MD techniques, the first ab initio molecular dynamics (AIMD) simulations, to investigate the temperature-dependent degradation dynamics of PFOA on $\gamma\text{-Al}_2\text{O}_3$ surfaces⁴⁹. According to DFT calculation, Gao et al. proposed the potential energy surface for possible reactions during the degradation of C6/C6 PFPiA in a discharge plasma system³³.*

DFT-based dynamics offers a valuable approach for studying the dynamic behavior of molecular systems with accurate electronic structure representation. However, given the substantial scale of our molecular system and the complex experimental conditions of our experimental conditions, a simulation method that combines ReaxFF MD and DFT appears to be better suited. In our

forthcoming research, we plan to utilize DFT-based dynamics methods to explore the decomposition mechanisms of contaminants. We appreciate your valuable suggestions. The suggested paper are cited as listed below:

References

33. Gao Z, et al. *Theoretical and experimental insights into the mechanisms of C6/C6 PFPiA degradation by dielectric barrier discharge plasma. J Hazard Mater* 424, 127522 (2022).
45. Biswas S, Yamijala S, Wong BM. *Degradation of Per- and Polyfluoroalkyl Substances with Hydrated Electrons: A New Mechanism from First-Principles Calculations. Environ Sci Technol* 56, 8167-8175 (2022).
46. de Souza BB, Hewage SA, Kewalramani JA, van Duin AC, Meegoda JN. *A ReaxFF-based molecular dynamics study of the destruction of PFAS due to ultrasound*. Environmental Pollution* 333, (2023).
47. Loganathan N, Wilson AK. *Adsorption, Structure, and Dynamics of Short- and Long-Chain PFAS Molecules in Kaolinite: Molecular-Level Insights. Environmental Science & Technology* 56, 8043-8052 (2022).
48. Fenton SE, et al. *Per- and Polyfluoroalkyl Substance Toxicity and Human Health Review: Current State of Knowledge and Strategies for Informing Future Research. Environmental Toxicology and Chemistry* 40, 606-630 (2021).
49. Biswas S, Wong BM. *Degradation of Perfluorooctanoic Acid on Aluminum Oxide Surfaces: New Mechanisms from Ab Initio Molecular Dynamics Simulations. Environmental Science & Technology* 57, 6695-6702 (2023).

Reviewer # 3

[Issue 1] Grammar. Overall, the grammar level is acceptable – but the authors often incorrectly use the word ‘the’, especially in their section headings. For example, ‘2.1 The experimental validation of the Reaxff model’ – in this case, ‘The’ should be removed – since ‘The’ indicates a uniqueness (like ‘The only experimental validation of the Reaxff model’. Note – also Reaxff should be spelled ReaxFF.

[Response] Accepted. Thank you for your careful comment. We have addressed the grammatical errors related to 'The' and 'ReaxFF', as well as other abused ‘The’. For example:

- **Line 112, P.5: 2.1** *(The) Experimental validation of (Reaxff) ReaxFF model*
- **Line 117, P.6:** *This section focused on (the) fluorine distribution in gaseous and solid products identified by ion chromatography (IC) (Fig.2).*

[Issue 2] I am concerned about the very high temperatures used in the ReaxFF simulations (up to 4000K). I appreciate that the authors seek to validate these temperatures by comparisons with experimental plasmas – but the experimental temperature equivalents are all deeply hidden in the SI material – the authors should at least mention these experimental temperatures in the main text. Do the authors consider these temperature realistic? At temperatures approaching 4000K electrons should be able to go to excited states – which is not described by ReaxFF.

[Response] Acknowledged. The OES measurement and fitting results of the temperature of plasma torch have been added into the Section 3.1.1 and Fig. 7.

- **Line 506, P.24:** *In order to determine the simulation condition of ReaxFF MD, the core temperature of plasma jet was measured by optical emission spectroscopy (OES) (Optosky ATP2400). The spectra optical emission spectra of plasma jet at different input power ($O_2/C = 0.93$) were shown in Fig. 7. Based on the comparison results of optical emission spectra of plasma jet with the standard spectrum, we determined that the core temperatures of the plasma jet at the input power of 22 kW, 24 kW, 26 kW, 28 kW, and 30 kW are 4500K, 4550K, 4650K, 4700K, and 4750K, respectively. Subsequently, a two-dimensional numerical model based on the DC plasma generator structure (Fig. S11), the core temperature, voltage, and*

current values of the plasma generator, was developed using Comsol software to simulate the temperature distribution of the plasma jet. As depicted in Figure S12, the gas temperature decreases rapidly as the distance from the center increases from 2mm to 6mm and remains relatively stable within the range of 6 to 10mm. Given that the plasma jet has a diameter of 10mm and the majority of the reaction zones occur at the periphery of plasma jet, we have determined the temperatures in the plasma reaction zone to be approximately 3300K, 3400K, 3600K, 3800K, and 4000K, corresponding to input powers of 22 kW, 24 kW, 26 kW, 28 kW, and 30 kW, respectively. Finally, the activated particle density of plasma jet was investigated by a zero-dimensional numerical modeling. The result of activated particle(s) density of plasma jet was shown in Fig. S13. More detailed measurements methods as well as simulation results on temperature and activated particle density of plasma jet were shown in Text S1, Table S4-S9 and Figure S11-S14.

Figure 7. The comparison between optical emission spectra of plasma jet and the standard spectrum under different input power (in the wavelengths range of 640nm-760nm): (a) power = 22 kW, O₂/C = 0.9, (b) power = 24 kW, O₂/C = 0.9, (c) power = 26 kW, O₂/C = 0.9, (d) power = 28 kW, O₂/C = 0.9, (e) power = 30 kW, O₂/C = 0.9.

Given that the torch temperatures were experimentally measured, we consider the obtained values to be reasonably reliable. As the reviewer correctly pointed out, at temperatures approaching 3000K or 4000K, electrons should be able to go to excited states, which is not described by ReaxFF MD. However, it's important to note that our study is centered on macromolecular solid reaction systems, with a primary focus on the formation and cleavage of chemical bonds between atoms and the rearrangement of atomic and molecular structures. Given the complexity and large scale of the reaction system in this research, the ReaxFF MD method is better suited to provide a chemically realistic representation of the reaction process at a meaningful scale. Furthermore, as shown in Fig. R1, the Pearson correlation coefficients (PCC) between defluorination efficiencies obtained from ReaxFF MD simulations and those observed in the plasma experimental system reached 96% in the mixed atmosphere, 64% in the oxygen atmosphere, 96% in the steam atmosphere, indicating the reliability of the ReaxFF MD simulation results in this research (Fig. R1).

Figure R1. The comparison between experimental inorganic fluorine ratios and MD simulated results under different (A) temperature ($O_2/C=2.7$, $H_2O/C=2.7$), (B) O_2/C ratio (temperature=3300K, $H_2O/C=0$), (C) H_2O/C ratio (temperature=3300K, $O_2/C=0$). The PCC represents the Pearson

correlation coefficients. The mineralized fluorine includes inorganic fluorides (HF, HOF, F-, F2) and fluorinated C1 compounds (such as COF₂, CF₃OH, FCOOH)

[Issue 3] Why was the ReaxFF simulation performed for 550 picoseconds? Was this because by that time all the chemistry had converged in the ReaxFF simulations – or was there a different reason to end the simulation at that time? 550 picoseconds is not a very long ReaxFF simulation – especially since the simulation boxes are relatively small.

[Response] Accepted. Thank you for your constructive comments. The duration of ReaxFF simulation was determined by analyzing the decay of the time autocorrelation function for temperature, press and total energy. The time autocorrelation functions, presented in exponential form, for the PTFE plasma gasification processes under oxygen, steam and mixed atmosphere have been added in Figure S14, S15 and S16, respectively. Descriptions relevant to the content have been included in Line 283-286 (Section 2.2.1), Line 366 – 369 (Section 2.2.2), Line 400 – 402 (Section 2.2.3) and Line 582-587 (Section 3.1.2). Given space constraints, only the time autocorrelation functions pertaining to oxygen atmosphere are shown below for reference. It could be observed from Figure S14 – S16 that the relaxation times (τ) of these autocorrelation functions for all variables in different reaction systems are below 100 ps. Moreover, all the autocorrelation functions drop to 0.001 within 500 ps. These findings suggest that all reaction systems attain equilibrium within 500 ps. Furthermore, given the suggestions, we extended the total simulation time to 1000 ps and re-simulated all the reaction systems in this study. In the revised manuscript, we have substituted Figure 2, Figure 3, Figure 5, Figure S5 – S10 with the new computational outcomes obtained from the 1000 ps simulation. Corresponding explanatory updates have been incorporated into Sections 2.2.1 - 2.2.3. Given space constraints, only the revised details pertaining to Section 2.2.1 and Figure 3 are shown below for reference.

- **Line 283-286, Section 2.2.1, P.13:** *The time autocorrelation functions for temperature, pressure, and total energy, presented in exponential form, were also computed for this process (Fig. S15(C)). Notably, all time autocorrelation functions dropped to 0.001 within 265 ps, verifying that the reaction system reaches equilibrium within 265 ps.*

Figure S15. Normalized time autocorrelation functions for the PTFE plasma gasification processes at O₂/C ratio of (A) 0.93, (B) 1.86, (C) 2.79, (D) 3.72, (E) 4.65 (temperature=3300K)

- **Line 288-293, Section 2.2.1, P.14:** The MD trajectory frames at O₂/C ratio of 2.79 in various stages were also consistent with the hypothesis related to the decomposition process (Fig. 3(C)) and suggested that the long chain PTFE molecules were degraded by plasma through pyrolysis (54ps) – oxidation and chain-shortening (58ps) – deep defluorination (75ps) – equilibrium (265

ps) (POCD) pathway in the presence of oxygen. At 1000 ps, the products were more dispersed, and most are C_1 compounds or inorganic fluorides.

Figure 3. MD simulation results of PTFE thermal plasma gasification process under oxygen atmosphere (O_2/C ratio=2.79, temperature=3300 K). (A) The evolution of products distribution with treatment time. (B) The main reaction pathways in (II) - (IV) stages. (C) Snapshots of MD trajectory frames at (a) 0 ps, (b) 54 ps, (c) 58 ps, (d) 75 ps, (e) 265 ps, (f) 1000 ps. The color codes for the atoms are red: C, violet: F, yellow: O.

[Issue 4] The force field used in this study – the Wood et al ReaxFF parameter set – may have issues with the C1-chemistry, in particular CO reactions with radicals. This was described and corrected by Ashraf and co-workers (Ashraf et al. J.Phys.Chem. A, 2017, 121, 1051-1068.) Can the authors discuss how the CO-chemistry issues, as described by Ashraf and co-workers, may affect their results?

[Response] Accepted. Thank you for your insightful comment. The correlation between experimental CO (15.66 - 18.28 wt.%) proportion and simulated CO proportion (12.26 – 30.26 wt.%) were weak (PCC=0.47). Moreover, some of reaction pathways involving fluorine-free carbon compounds (CO, CO₂, C₁₋₃) appeared to be less reasonable. These phenomena may be related to the limitations of the force field parameters utilized in this study, which might not precisely capture the behavior of C1 chemistry with sufficient precision. These potential effects have been added in the Section 2.1 and Section 2.2.2 of the revision manuscript.

- **Line 168-172, Section 2.1, P.8:** However, the correlation between experimental CO (15.66 - 18.28 wt.%) proportion and simulated CO proportion (12.26 – 30.26 wt.%) were weak (PCC=0.47). This phenomenon may be attributed to the limitations of the force field parameters utilized in this study, which might not precisely capture the behavior of C1 chemistry with sufficient precision⁶⁰.
- **Line 362-366, Section 2.2.2, P.17:** *It is worth noting that while the combination of ReaxFF MD and DFT simulations in this section offers a reasonable explanation for the defluorination process of micromolecular perfluorocarbons, the conversion pathways involving fluorine-free carbon compounds (CO, CO₂, C₁₋₃) were complex and still not fully understood. Further investigation of the detailed reaction mechanisms involved in small hydrocarbons may require the use of more precise and targeted CHO force field parameters⁶⁰.*

However, it is worth mentioning that evaluating the applicability of force field parameters requires a comparison with the sufficient dataset of experimental and theoretical data, such as molecular structure, vibration frequency, reaction heat, and activation energy. Due to the limitations of our theoretical foundation and computing resources, it is challenging to achieve an accurate and quantitative validation of the performance of the force field parameters developed by Wood et al.,

which is beyond the scope of this study. Moreover, in this study, the force field set developed by Wood et. al provides a reliable prediction of the defluorination efficiency and the mineralization mechanism of PTFE. Additionally, this force field parameter set can produce HCO radicals, which deviates from the mentioned study (J.Phys.Chem. A, 2017, 121, 1051-1068). Thus, the above discussion are speculations rather than quantitative results.

References:

60. Ashraf C, van Duin ACT. *Extension of the ReaxFF Combustion Force Field toward Syngas Combustion and Initial Oxidation Kinetics. The Journal of Physical Chemistry A* 121, 1051-1068 (2017).

Reviewer # 4

[Issue 1] The reviewer believes that the term 'global optimization' on Line 90 should be used very carefully in studies involving computational approaches. Both DFT and ReaxFF MD simulations have their own limitations as models and solvers, and we cannot guarantee that the obtained pathways and the associated energies are placed at the global minima. The author's intention in this sentence is probably to emphasize that the operation condition to maximize the effect of C-F scissoring was found from simulation. Although extensive computational results have been performed, the cases are still within a finite compositional range, so it is a leap to discuss the findings as the result of 'global optimization' of the experimental conditions.

[Response] Accepted. Thanks for your constructive comment. As highlighted by the reviewer, neither DFT nor ReaxFF MD could consistently converge to the global minimum energy state of the system. Consequently, all terms involving 'global optimization' have been revised as follows:

- **Line 32, P.2:** *under the ~~(global-optimized-conditions)~~ operation conditions optimized through response surface methodology*
- **Line 97-101, P.5:** *Moreover, the ~~(global)~~ optimal combination of operation conditions was obtained to maximize the turnover rate of PTFE towards inorganic fluorine through ReaxFF MD and response surface methodology (RSM)*
- **Line 459, P.21:** *The MD simulations were designed and regression analyzed through Box-Behnken design in the Design-Expert software to explore the synergic influences of operational conditions and found the ~~(global)~~ maximized values of IF and MR*

[Issue 2] The authors compared the ReaxFF simulation results of the distribution of fluorocarbon decomposition species in three different environments with the experimental IC results. The reviewer basically agrees with the authors' assertion that the results in Figure 2 show that the qualitative trends of experiments and simulations are consistent. However, there is a logical leap to conclude that the validity of the simulation model itself has been verified based on these results. Above all, only one optimal condition found from the proposed RSM model has been experimentally verified for thermal decomposition of PTFE, so it cannot be said that the value is an 'experimentally optimal condition'. In other words, the RSM model

established from the multiscale simulation finds one of the appropriate improvement conditions under the given conditions, not the global optimum condition of the thermal plasma process. Therefore, the abstract, introduction, and title that demonstrated that an appropriate operation condition was found from simulation conditions should all be modified.

[Response] Accepted. As pointed by the reviewer, the conditions optimized through multiscale simulations could not represent the globally optimal conditions observed in experiments, thus the abstract and introduction have been revised accordingly. The words 'experimental validation' in the title actually refers to the investigation of mechanisms rather than the optimization of operational conditions. To keep the title's simplicity no specification is added in title.

- **Line 31, P.2:** *under the ~~(global optimized conditions)~~ operation conditions optimized through response surface methodology*
- **Line 97-101, P.5:** *Moreover, the ~~(global)~~ optimal combination of operation conditions was obtained to maximize the turnover rate of PTFE towards inorganic fluorine through ReaxFF MD and response surface methodology (RSM), ~~which is supported by the experimental results.~~ Experimental results have demonstrated that this combination of operation conditions could effectively enhance the defluorination efficiency of the PTFE plasma gasification process.*

In order to make the validation more comprehensive, a further comparison has been plotted as shown in Figure R3. The results show that the Pearson correlation coefficients (PCC) between defluorination efficiencies obtained from ReaxFF MD simulations and those observed in the plasma experimental system reached 96% in the mixed atmosphere, 64% in the oxygen atmosphere, 96% in the steam atmosphere. A lower PCC of 64% is observed in the oxygen atmosphere while still acceptable for such scale modeling. Overall, we think the accuracy is enough for the validation.

Figure R3. The comparison between experimental inorganic fluorine ratios and MD simulated results under different (A) temperature ($O_2/C=2.7$, $H_2O/C=2.7$), (B) O_2/C ratio (temperature=3300K, $H_2O/C=0$), (C) H_2O/C ratio (temperature=3300K, $O_2/C=0$). The PCC represents the Pearson correlation coefficients. The mineralized fluorine includes inorganic fluorides (HF, HOF, F-, F₂) and fluorinated C1 compounds (such as COF₂, CF₃OH, FCOOH)

[Issue 3] Continuing from the comments immediately above, the authors arbitrarily mix qualitative and quantitative comparisons when discussing the differences between the experimental and simulated results presented in Figure 2. For example, the author argued that COF₂ and dot_CFO₂ appearing in ReaxFF are the result of not considering the reaction with water. If so, does each reaction have such a self-evident chemical pathway that the experiment (or simulation considering the reaction with water) yields a reaction rate close to 100%? It is necessary to explain, at least qualitatively, what state the COF₂ and dot_CFO₂ products will remain in the real environment. Similarly, CF₃O occupies a significant proportion in the MD simulation, whereas CF₄ occupies a similar proportion in the experiment. No discussion of this can be found in the main text. A similar issue is raised with the dot_CFO₂ results in Figure 2A.

Is the contribution of dot_CFO2 in the MD simulation completely explained by the CO2 in the experiment?

As far as the reviewer understands, the results in Figure 2 are the only part in this manuscript in which direct comparisons with experimental data have been made, and are therefore of great importance. Nevertheless, in the present manuscript, the reasons for the discrepancies between the data obtained from the simulation and the experimental values are limited to listing. The reviewer asks that the authors specify precisely in the text what part of the results observed experimentally this simulation succeeded in reproducing and what findings were captured accordingly.

[Response] Accepted. The reaction pathway of $\cdot\text{CFO}_2$ to produce CO_2 ($\cdot\text{CFO}_2 \rightarrow \cdot\text{F} + \text{CO}_2$), COF_2 to produce CO_2 and HF ($\text{H}_2\text{O} + \text{COF}_2 \rightarrow \text{CO}_2 + 2\text{HF}$), as well as the reaction of CF_3O to produce CF_4 ($\text{CF}_3\text{O} + \cdot\text{F} \rightarrow \text{CF}_4 + \text{O}\cdot$) have been demonstrated by previous researches. Corresponding references have been added in the Section 2.1 (Reference 50-59). To precisely identify the portion of experimentally observed results successfully reproduced by this simulation, we applied statistical methods to assess the reliability of the simulation results. First, we calculated the Pearson correlation coefficients to assess the relationship between simulated and experimental defluorination efficiencies (refer to Fig. 2). The PCC value reached 96% in the steam atmosphere, 64% in the oxygen atmosphere, 96% in the mixed atmosphere. These high PCC values indicate a robust linear association between the simulated and experimental defluorination efficiencies. Second, we calculated the Pearson correlation coefficients to examine the correspondence between the experimental proportion of final products (CO_2 , CF_4) and the simulated proportions of intermediates ($\cdot\text{CFO}_2$, COF_2 and CF_3O) (refer to Fig. 2). The PCC values of main gaseous product proportion reached 0.68 in the mixed atmosphere, 0.67 - 0.82 in the oxygen atmosphere and 0.43-0.93 in the steam atmosphere. These results demonstrated the strong positive correlation between the simulated and experimental results regarding the main gaseous products. The above results have been included in the Figure 2 and corresponding explanatory revisions have been included in Sections 2.1.

- **Line 134, P.6:** *Conversely, the inorganic fluorine ratios derived from experiments and the MD simulations under the oxygen atmosphere and MD simulated values under the oxygen atmosphere showed significant differences and lower PCC values (Fig.2B). These*

disparities can be attributed to the high proportion of COF_2 and $\cdot\text{CFO}_2$ in the products (Fig.2C). These compounds can react with the water in the sampling absorbent liquid ($\cdot\text{CFO}_2 \rightarrow \cdot\text{F} + \text{CO}_2$; $\text{H}_2\text{O} + \text{COF}_2 \rightarrow \text{CO}_2 + 2\text{HF}$), resulting in the formation of CO_2 and HF and the increased inorganic fluorine contents in the actual experiment system^{50, 51, 52, 53, 54, 55, 56, 57}. The higher PCC (0.64) between experimental inorganic fluorine ratio and simulated mineralized fluorine ratio (including fluorinated Cl compounds and inorganic fluorine) supported this explanation (Fig.S1).

- Line 151, P.7:** The experimental dominant gaseous products were CO_2 and CO (86.99 - 90.20 wt.%) in the mixed atmosphere, while the simulated dominant gas component were CO_2 and $\cdot\text{CFO}_2$ (81.61 - 91.32 wt.%) (Fig. 2A). This difference could be explained that the $\cdot\text{CFO}_2$ intermediate is unstable and will react with steam to form inorganic fluorides and CO_2 in the actual mixed-atmosphere gasification system ($\cdot\text{CFO}_2 \rightarrow \cdot\text{F} + \text{CO}_2$)^{50, 51, 52, 53, 54, 55}. The PCC value exceeding 0.60 between the experimental CO_2 proportion and the simulated sum proportion of CO_2 and $\cdot\text{CFO}_2$ substantiated this hypothesis (Fig.2A). The experimental predominant gas components were COF_2 and CO_2 (70.35 - 81.06 wt.%) in the oxygen atmosphere, while the simulated predominant components were COF_2 and $\cdot\text{CFO}_2$ (44.87 wt.% - 59.90 wt.%) (Fig. 2B). This difference was also attributed to the conversion of $\cdot\text{CFO}_2$ in the actual oxygen plasma gasification process. This interpretation was supported by the observation that the PCC value of 0.67 between experimental CO_2 proportion and the simulated $\cdot\text{CFO}_2$ proportion, demonstrating a significant correlation (Fig. 2B). Moreover, the $\text{CF}_3\text{O}\cdot$ intermediate could be converted into CF_4 in the actual oxygen plasma gasification process^{58,59}, which ultimately led to the higher CF_4 content in experimental results. This explanation was substantiated by a PCC value of 0.77. As regards the steam plasma gasification process, the experimental and simulated dominant gaseous products were both CO_2 . Furthermore, the PCC value between experimental CO_2 proportion and simulated value under the steam gasification reached up to 0.93, exhibiting their high consistency (Fig. 2C).

Figure 2. The comparison of defluorination efficiency (I) and gaseous products composition (II) between experiment values and ReaxFF MD simulation values under different (A) temperature ($O_2/C=2.7$, $H_2O/C=2.7$), (B) O_2/C ratio (temperature=3300K, $H_2O/C=0$), (C) H_2O/C ratio (temperature=3300K, $O_2/C=0$). The PCC represents the Pearson correlation coefficients. The error bar represents the standard deviation (SD).

References:

50. *Kechoindi S, Ben Yaghlane S, Terzi N, Palaudoux J, Hochlaf M. Characterization and photochemistry of XCO₂ (X = F, NH₂, CH₃) radicals. The European Physical Journal Special Topics 232, 1905-1916 (2023).*
51. *Arguello GA, Grothe H, Kronberg M, Willner H, Mack H-G. IR and Visible Absorption Spectrum of the Fluoroformyloxyl Radical, FCO₂.bul., Isolated in Inert Gas Matrixes. The Journal of Physical Chemistry 99, 17525-17531 (1995).*
52. *Berasategui M, Burgos Paci MA, Argüello GA. Properties and Thermal Decomposition of the Hydro-Fluoro-Peroxide CH₃OC(O)OOC(O)F. The Journal of Physical Chemistry A 118, 2167-2175 (2014).*
53. *Kovács A, Konings RJM, Gibson JK, Infante I, Gagliardi L. Quantum Chemical Calculations and Experimental Investigations of Molecular Actinide Oxides. Chemical Reviews 115, 1725-1759 (2015).*
54. *Arnold DW, Bradforth SE, Kim EH, Neumark DM. Study of halogen-carbon dioxide clusters and the fluoroforomyloxyl radical by photodetachment of X-(CO₂) (X=I,Cl,Br) and FCO-2. The Journal of Chemical Physics 102, 3493-3509 (1995).*
55. *Burgess Jr D, Zachariah MR, Tsang W, Westmoreland PRJPIE, Science C. Thermochemical and chemical kinetic data for fluorinated hydrocarbons. 21, 453-529 (1995).*
56. *Farlow MW, Man EH, Tullock CW, Richardson RD. Carbonyl Fluoride. In: Inorganic Syntheses (1960).*
57. *Francisco JS. A study of the gas-phase reaction of carbonyl fluoride with water. Journal of Atmospheric Chemistry 16, 285-292 (1993).*
58. *Nguyen D, Lee WJRa. Effects of ambient gas on cold atmospheric plasma discharge in the decomposition of trifluoromethane. 6, 26505-26513 (2016).*
59. *Yamamoto M, Yamashita K, Sadakata M. Study of the Reactions of O- with CF₄ and CHF₃ by Ab Initio Calculations. Bulletin of The Chemical Society of Japan - BULL CHEM SOC JPN 75, 1483-1491 (2002).*

[Issue 4] Regarding the reactive MD simulations, the reviewer concerns that the overall time integration applied to the MD models is too short. In particular, it is necessary to secure additional validity for the simulation methodology that causes thermal decomposition by raising the temperature of the model. The addition of supplementary simulation results is needed to address the following questions.

[Response] Acknowledged. Special thanks for your constructive comment. In response to this suggestions, we extended the total simulation time to 1000 ps and re-simulated all the reaction systems in this study. In the revised manuscript, we have substituted Figure 2, Figure 3, Figure 5, Figure S5 – S10 with the new computational outcomes obtained from the 1000 ps simulation. Corresponding explanatory updates have been incorporated into Sections 2.2.1 - 2.2.3. Given space

constraints, only the revised details pertaining to Section 2.2.1 and Figure 3 are shown below for reference. In addition, we calculated the time autocorrelation functions, presented in exponential form, for the PTFE plasma gasification processes under oxygen, steam and mixed atmosphere. This analysis aimed to determine whether the reaction systems had reached equilibrium, and the results are shown in Figure S14, S15, and S16. It could be observed from Figure S14 – S16 that the relaxation times (τ) of these autocorrelation functions for all variables in different reaction systems are below 100 ps. Moreover, all the autocorrelation functions drop to 0.001 within 500 ps. These findings suggest that all reaction systems attain equilibrium within 500 ps.

- **Line 288-293, Section 2.2.1, P.14:** *The MD trajectory frames at O_2/C ratio of 2.79 in various stages were also consistent with the hypothesis related to the decomposition process (Fig. 3(C)) and suggested that the long chain PTFE molecules were degraded by plasma through pyrolysis (54ps) – oxidation and chain-shortening (58ps) – deep defluorination (75ps) – equilibrium (265 ps) (POCD) pathway in the presence of oxygen. At 1000 ps, the products were more dispersed, and most are C_1 compounds or inorganic fluorides.*

For the following reasons, we find it unnecessary to conduct additional temperature elevation simulations to secure additional validity for the simulation methodology. It can be observed from Figure 6(b) and Figure S10 that the MR value, IF value, and the evolution trend of product distribution exhibit low sensitivity to temperature variations. In addition, the temperatures employed in the ReaxFF MD simulations were measured using optical emission spectroscopy (OES), indicating the consistency between the simulation and actual experimental temperature conditions. Furthermore, the excessively elevated temperatures can result in greater electron excitations, potentially increasing the discrepancy between simulated and experimental results. Therefore, we believe that the simulated temperature conditions and thermal behavior of PTFE in this study are highly representative of the actual plasma gasification process.

Figure 3. MD simulation results of PTFE thermal plasma gasification process under oxygen atmosphere (O_2/C ratio=2.79, temperature=3300 K). (A) The evolution of products distribution with treatment time. (B) The main reaction pathways in (II) - (IV) stages. (C) Snapshots of MD trajectory frames at (a) 0 ps, (b) 54 ps, (c) 58 ps, (d) 75 ps, (e) 265 ps, (f) 1000 ps. The color codes for the atoms are red: C, violet: F, yellow: O.

[Issue 5] - How was the initial density of the presented model determined? In the MD recipe presented in the manuscript, there is no process to converge the density of a given PTFE system.

[Response] Acknowledged. The process of determining the initial density is as follows. First, the initial simulation units were established by the Amorphous Cell module in Material Studio software (MS). Then, the geometric optimization of periodic unit was carried out through Dmol3 module in MS. After that, the NPT ensemble of Forcite module in Material studio was used to relax the initial structure and get the initial density of simulation units. The density evolution of the representative reaction units was shown in Fig. R4.

Figure R4. The density relaxation process of the simulation units under (a) Oxygen atmosphere (O_2/C ratio=2.79, temperature=3300K); (b) Steam atmosphere (H_2O/C ratio=2.79, temperature=3300K)

Nevertheless, if the simulation is conducted utilizing post-relaxation densities, it will lead to the following adverse consequences. In an oxygen atmosphere, the low density of the gasification agent led to a reduced collision efficiency between PTFE molecules and oxygen. As a result, it required 2-3 nanoseconds for the reaction systems to reach equilibrium. In the steam atmosphere, there exists inadequate space for hydrogen bonding, resulting in the formation of anomalous structures and associated issues.

Therefore, the initial density of reaction units was set to 1g/cm^3 . On the one hand, the pressure-accelerated dynamics under the oxygen atmosphere could promote the effective collision of molecules and save the computational time and sources (*Powder Technology* 410, 117837 (2022); *ChemSystemsChem* 2, e1900043 (2020)).

On the other hand, the slightly lower pressure of the simulation unit under the steam atmosphere could provide more suitable conditions for hydrogen bond formation. The comparative analysis of simulation results utilizing post-relaxation density and a fixed 1 g/cm³ density is presented in the Figure R5-R6. It can be observed that apart from the start times of each stage, the primary reaction processes and pathways exhibit remarkable similarity.

Figure R5. The comparison of MD results regarding the evolution of product distribution during the PTFE thermal plasma gasification process under an oxygen atmosphere (O_2/C ratio = 2.79,

temperature = 3300K): (A) utilizing the post-relaxation density as the initial density and (B) using a fixed initial density of 1 g/cm³.

Figure R6. The comparison of MD results regarding the evolution of product distribution during the PTFE thermal plasma gasification process under a steam atmosphere (H_2O/C ratio = 2.79, temperature = 3300K): (A) utilizing the post-relaxation density as the initial density and (B) using a fixed initial density of 1 g/cm³.

[Issue 6] Are the prepared models capable of smooth temperature control by Langevin thermostats until reaching a level of 4000 K? Present the temperature profile provided as input during the thermal decomposition simulation and the resulting total energy and the temperature change profiles of the system.

[Response] Accepted. We sincerely appreciate your diligence in reviewing our work and apologize for the oversight in incorrectly specifying the temperature control method. It has been corrected to "Nose-Hoover thermostats" in the revised manuscript (Line 576, Page 27). The temperature and total energy evolution of the reaction systems in oxygen, steam, and mixed atmospheres have been added into the revised manuscript, as shown in Figure S18-S20. Given space constraints, only the temperature and total energy profiles pertaining to oxygen atmosphere are shown below for reference.

Figure S18. The temperature and total energy profiles for the PTFE plasma gasification processes at the O_2/C ratio of (A) 0.93, (B) 1.86, (C) 2.79, (D) 3.72, (E) 4.65 (temperature=3300K)

[Issue 7] In the opinion of reviewers, 50K/ps is too harsh and too fast a heating condition. This can cause some important events of the system's pyrolysis species to be missed or overestimated (such as the recombination of temporarily broken molecules). For a representative model, please prove with data that the considered heating rate condition is slow enough to calculate the profile of the degraded species ratio and therefore exhibits high convergence.

[Response] Accepted. In response to the this suggestion, we conducted MD supplementary simulations of the PTFE plasma gasification process under both oxygen and steam atmospheres, utilizing heating rates of 10K/ps, 30K/ps, and 50K/ps.

The detailed evolution of product composition under an oxygen atmosphere with different heating rates were shown in the Fig. R7. It can be observed that the reaction process exhibits similarity under different heating rates and can be categorized into five stages: I-heating, II-chain initiation (pyrolysis), III-chain transfer (oxidation and chain-shortening), IV-chain termination (deep defluorination), and V-equilibrium. The decrease of heating ratio could extend the duration of the heating, chain initiation and chain transfer stages, allowing some reaction pathways to become clearer, such as the reaction of perfluorocarbon radicals with $^1\text{O}_2/\text{O}$ radicals to generate peroxy radicals. Nevertheless, during the main five reaction stages, the proportions and variation trends of degraded species in reaction systems with varying heating rates exhibit significant similarity, indicating that the reaction pathways and mechanisms utilizing heating rates of 50K/ps are representative. Moreover, the Pearson correlation coefficients (PCC) between experimental inorganic fluorine ratios and MD simulated results utilizing heating rates of 10K/ps, 30K/ps, and 50K/ps were 0.52, 0.30 and 0.64 (Fig.R8), showing a stronger correlation between the experimental results and the MD model under the heating rate of 50K/ps. This phenomenon may be attributed to the instantaneous extremely high temperature of the plasma torch jet without preheating.

As shown in Fig.R9, the detailed evolutions of product composition under a steam atmosphere with different heating rates demonstrate the similarity and can be divided into four stages: I-chain initiation (F abstraction), II-chain transfer (hydrolysis), III-chain termination (deep defluorination), and IV-equilibrium. The decrease of heating ratio could extend the duration of the chain initiation and chain transfer stages, allowing some reaction pathways to become clearer, such as the release of $:\text{CF}_2$ radicals. However, throughout the main four reaction stages, the proportions and trends of

degraded species in reaction systems with varying heating rates are close, suggesting that the reaction pathways and mechanisms at 50K/ps are representative. Furthermore, the PCC between experimental inorganic fluorine ratios and MD simulated results utilizing different heating rates are all beyond 0.90, showing a high correspondence between the MD models and experimental results (Fig.R10). Taking into account both the MD simulation models for oxygen and steam atmospheres, we find it reasonable to utilize a heating rate of 50K/ps for simulating the actual plasma gasification process in this study.

Figure R7. The comparison of MD results regarding the evolution of product distribution during the PTFE thermal plasma gasification process under an oxygen atmosphere (O_2/C ratio = 2.79, temperature = 3300K), utilizing heating rates of (A) 10K/ps, (B) 30K/ps, (C) 50K/ps.

Figure R8. The comparison between experimental inorganic fluorine ratios and MD simulated mineralized fluorine ratio under the oxygen atmosphere (temperature=3300K), utilizing heating rates of (A)10K/ps, (B) 30K/ps, (C) 50K/ps. The PCC represents the Pearson correlation coefficients. The mineralized fluorine includes inorganic fluorides (HF, HOF, F, F₂) and fluorinated Cl compounds (such as COF₂, CF₃OH, FCOOH).

Figure R9. The comparison of MD results regarding the evolution of product distribution during the PTFE thermal plasma gasification process under a steam atmosphere (H_2O/C ratio = 2.79, temperature = 3300K), utilizing heating rates of (A) 10K/ps, (B) 30K/ps, (C) 50K/ps.

Figure R10. The comparison between experimental fluorine mineralization ratios and ReaxFF molecular dynamics (MD) simulation results under the steam atmosphere (temperature=3300K), utilizing heating rates of (A)10K/ps, (B) 30K/ps, (C) 50K/ps. The PCC represents the Pearson correlation coefficients.

[Issue 8] As stated by the authors, the molecular weight of PTFE covered by the MD simulations is clearly too small for the experimental conditions. Additional data is needed to justify that the effect of molecular weight differences does not need to be carefully considered in this study. One solution seems to be to model an infinite length structure of self-connected PTFE chains beyond periodic boundary conditions and perform thermal decomposition simulations.

[Response] Acknowledged. Simulating an infinite length structure of PTFE chains may require specialized techniques, such as asymmetric periodic boundary conditions, to mitigate the impact of boundary effects on simulation results. However, this approach can introduce a series of issues, including the difficulty of guaranteeing that the O₂/C and H₂O/C ratio aligns with experimental conditions and the topological constraints at the junction of mirrored spaces, which may hinder the PTFE chain from undergoing breaking during thermal degradation process.

In response to reviewer's comments, we performed MD supplementary simulations of the PTFE plasma gasification process with the polymerization degrees of 30 (C₆₀F₁₂₂), 270 (C₅₄₀F₁₀₈₂), and 510 (C₁₀₂₀F₂₀₄₂), under both oxygen and steam atmospheres. The detailed evolution of product composition under an oxygen atmosphere with different polymerization degrees were shown in the Fig. R11. It can be observed that the main reaction process displays similarities across different polymerization degrees and can be divided into five stages: I-heating, II-chain initiation (pyrolysis), III-chain transfer (oxidation and chain-shortening), and IV-chain termination (deep defluorination), and V-equilibrium. Notably, with an increase in the degree of polymerization, the reaction system requires more time to transition from the chain transfer stage to the chain termination stage. This result mainly stems from the fact that with an increase in the degree of polymerization among the reactants, the macromolecular perfluorocarbon intermediates require more frequent release of small molecules (e.g. C₂F₄, CF₂O₂, and COF₂) to gradually shorten the carbon chain. Correspondingly, the number of PFAS(C₂₋₃) in the carbon transfer stage also gradually increases. Nevertheless, during the main five reaction stages, the proportions and variation trends of macromolecular perfluorocarbons (PFAS(C₄₊)), micromolecular perfluorocarbons (PFAS(C₁₋₃)), and inorganic fluorides across different polymerization degrees show significant similarity. This result demonstrates that the reaction pathways and mechanisms observed at the polymerization degree of

30 are representative. Moreover, the Pearson correlation coefficients (PCC) between experimental inorganic fluorine ratios and MD simulated results with the polymerization degrees of 30, 270, and 510 were 0.64, 0.63 and 0.59 (Fig.R12), respectively, showing a stronger correlation between the experimental results and the MD model at the polymerization degree of 30. This phenomenon may be explained by the enhanced boundary effects and the increased prominence of randomness in sampling of the larger simulation boxes.

As depicted in Fig.R13, the detailed evolutions of product composition under the steam atmosphere, considering various polymerization degrees, exhibit remarkable similarity and can be categorized into four stages: I-chain initiation (F abstraction), II-chain transfer (hydrolysis), III-chain termination (deep defluorination), and IV-equilibrium. It could be observed that the increase of polymerization degrees could extend the duration of the chain termination stages. This phenomenon may be owing to the increased number of micromolecular perfluorocarbon intermediates and the accelerated occurrence of reactions including HF elimination, F abstraction and radical substitution reaction, as the degree of polymerization increases. However, during the main four reaction stages, the proportions and trends of degraded species in reaction systems with varying polymerization degrees are close, suggesting that the reaction pathways and mechanisms at the polymerization degree of 30 are representative. Furthermore, the PCC between experimental inorganic fluorine ratios and MD simulated results utilizing different heating rates are all beyond 0.95, showing a high correspondence between the MD models and experimental results (Fig.R10). In considering the fitting performance of MD models under both oxygen and steam atmospheres, as well as computational resource constraints, we find it reasonable to select the polymerization degree of 30 for simulating the actual plasma gasification process in this study.

Figure R11. The comparison of MD results regarding the evolution of product distribution during the PTFE thermal plasma gasification process under an oxygen atmosphere (O_2/C ratio = 2.79, temperature = 3300K), with the polymerization degrees of (A) 30 ($C_{60}F_{122}$), (B) 270 ($C_{540}F_{1082}$), and (C) 510 ($C_{1020}F_{2042}$).

Figure R12. The comparison between experimental inorganic fluorine ratios and MD simulated mineralized fluorine ratio under the oxygen atmosphere (temperature=3300K), with the polymerization degrees of (A) 30 ($C_{60}F_{122}$), (B) 270 ($C_{540}F_{1082}$), and (C) 510 ($C_{1020}F_{2042}$). The PCC represents the Pearson correlation coefficients. The mineralized fluorine includes inorganic fluorides (HF, HOF, F, F₂) and fluorinated C1 compounds (such as COF₂, CF₃OH, FCOOH).

Figure R13. The comparison of MD results regarding the evolution of product distribution during the PTFE thermal plasma gasification process under a steam atmosphere (H_2O/C ratio = 2.79, temperature = 3300K), with the polymerization degrees of (A) 30 ($C_{60}F_{122}$), (B) 270 ($C_{540}F_{1082}$), and (C) 510 ($C_{1020}F_{2042}$).

Figure R14. The comparison between experimental inorganic fluorine ratios and MD simulated mineralized fluorine ratio under the steam atmosphere (temperature=3300K), with the polymerization degrees of (A) 30 ($C_{60}F_{122}$), (B) 270 ($C_{540}F_{1082}$), and (C) 510 ($C_{1020}F_{2042}$). The PCC represents the Pearson correlation coefficients.

[Issue 9] The supplementary file contains a lot of discussion and related background on auxiliary results. However, the organization of the text is too incomplete, and each sentence is not easily read. Although all of this is supplementary material, a minimum level of readability should be ensured.

[Response] Accepted. Thank you for your thorough review. We have had this supplementary material revised by a native speaker to ensure its readability. Here are a few examples of the implemented revisions:

- **Line 919, P.42:** *Before conducting the PTFE thermal plasma degradation experiment (~~the PTFE thermal plasma degradation experiment~~), the gas temperature of plasma jet in(~~at~~) the presence of oxygen and steam were measured using(~~by~~) optical emission spectroscopy (Optosky ATP2400).*
- **Line 921, P.42:** *The emission spectra of the plasma torch (~~spectrum of the plasma torch emission~~) under different reaction conditions were recorded by using an optical fiber(~~fibre~~) located 5cm away from the plasma generator.*
- **Line 956, P.45:** *As shown in Fig S10, the gas temperature is mainly determined by the input power and exhibits low sensitivity (~~scarcely influenced by~~) to variations in the gas agent concentrations (O_2 or H_2O).*
- **Line 958, P.45:** *This observation can be attributed to the limited involvement of oxygen and steam in the plasma generation process. (~~This result may be explained by that the oxygen and steam didn't participate in the plasma generation~~).*
- **Line 1021, P.50:** *This model sets the electron and gas temperature based on measurements obtained from the spectrum. (~~The electron temperature and the gas temperature are set as the temperature measured by the spectrum in this model~~).*

[Issue 10] What does 'Operational variations' on Line 53 mean? Does that phrase refer to process conditions that deviate from the optimized incineration process mentioned in the previous sentence on Line 49? If so, why should we consider such variations? Or, is this sentence intended to emphasize that the risk of yield degradation is too high for even slight deviations from optimized conditions? The intention is not clearly understood only with the

contents presented in the manuscript, so it seems that the contents need to be specified or modified.

[Response] Accepted. The phrase 'Operational variations' refers to the optimal process conditions mentioned in the previous sentence. This statement emphasizes that due to the excellent stability of C-F bond in fluoropolymers, the deviations in thermochemical operation conditions, such as temperatures below 870 °C or residence times below 4s, may lead to the reintroduction of hazardous perfluorocarbons (PFCs). In the revised manuscript, this sentence has been revised:

- **Line 53, P.3:** *However, due to the excellent stability of C-F bond in fluoropolymers, the variations in operation conditions will reduce defluorination performance, reintroduce hazardous perfluorocarbons (PFCs) and perfluorinated carboxylic acids (PFCAs) (C3-C14) into the environment.*

[Issue 11] The sentence placed on Lines 57 to 60 seems to be about the difficulty of quantitative detection of the thermal decomposition of PTFE. However, it does not seem to have much to do with the risk of byproduct or mineralization effectiveness. In other words, this sentence interrupts the flow of content before and after.

[Response] Acknowledged. This sentence aims to emphasize that due to the challenges associated with quantitatively detecting all fluorine components, there may be unknown fluorine by-products, the risks of which remain unclear. Considering the suggestions, we have revised this sentence as follows:

- **Line 56, P.3:** *Moreover, most of publications thus far fall short in quantitatively accounting for all fluorinated products in the thermal treatment system and achieving a complete fluorine mass balance, due to the detection limitation of conventional experiment methods. These limitations make the researchers question the effectiveness of PTFE mineralization via thermochemical conversion technologies and the associated risks of fluorochemical byproducts.*

[Issue 12] The order of text in the Methods section should follow the workflow presented in Figure 1A. From the reader's point of view, the present manuscript first encounters validation

of the main data (Figure 2) and experiments without fully understanding the modeling environment and conditions of the ReaxFF simulation. Because of this, the reviewer felt that the manuscript was less readable.

[Response] Acknowledged. The order of text in the Methods section is generally in line with the workflow depicted in Figure 1A. Section 3.1.1 is dedicated to establishing the initial simulation conditions using OES experimental approaches, corresponding to step 1 in Fig. 1A. Sections 3.1.2 to 3.1.4 elaborate on the ReaxFF MD, DFT, and RSM simulation methods in detail, corresponding to steps 2, 3, and 5, respectively, in Fig. 1A. Subsequently, Section 3.2 focuses on presenting the experimental validation method, corresponding to step 4 in Fig. 1A. Positioning the RSM section (3.1.4) before experimental verification methodology (3.2), rather than at the end, was intended to offer a thorough introduction to multi-scale models. In addition, the interleaved introduction of experimental methods and simulation methods may cause potential confusion for readers. Considering the comment, the heads of Section 3.1.1 and 3.2 have been revised as follows to make the logic of Method clearer:

- **Line 503, P.24:** *3.1.1 Establishing Initial Simulation Parameters: Measurement of Plasma Characteristics*
- **Line 614, P.29:** *3.2 Experimental validation methodology*

There are two reasons for placing the experimental validation in Section 2.1. First, according to the guidelines of *Nature Communications*, the Results and Discussions section should precede the Methods section. Second, the model validation part serves as the foundation for subsequent investigations into the thermal degradation mechanism and the influences of operational parameters. By confirming the model's reliability through validation, the ensuing results and discussions gain credibility.

[Issue 13] Figures 1B-E should be drawn separately from Figure 1A. These data are related to Section 3.1 and are not suitable for presentation along with the modeling overview in the introduction.

[Response] Accepted. Figures 1B-E have been excluded from Figure 1. The updated Figure 1 is displayed below:

Figure 1. Multiscale modeling framework for exploring the PTFE degradation performance and mechanisms by DC thermal plasma.

[Issue 14] Too many typos related to abbreviations and jargon (PFTE -> PTFE (Line 50); Molecular dynamics -> Molecular dynamics (Line 87); Reaxff or ReaxFF? (Line 106); The origin column -> the orange (?) column (Line 132). Also, it looks like the colors referring to COF₂ and dot_CFO₂ in the text are reversed.). Typos throughout the manuscript must be corrected before entering the next round.

[Response] Accepted. Thanks for your careful comment. We have thoroughly reviewed and correct the similar typos. The Figure 2 have been revised with CFO₂ depicted as a bar filled with diagonal slashes (refer to Fig.2).

- **Line 50, P.3:** *3.1.1 Establishing Initial Simulation Parameters: Measurement of Plasma Characteristics*
- **Line 100, P.5:** *Global optimized operation conditions were obtained to maximize the turnover rate of PTFE towards inorganic fluorine through ReaxFF MD and response surface methodology (RSM)*

- **Line 117, P.6: 2.1** *(The) Experimental validation of ~~(Reaxff)~~ ReaxFF model*
- **Line 459, P. 21:** *The average IF value at O₂/C ratio of 4.65, H₂O/C ratio of 4.00 and input power of 22 ~~(kW)~~ kW was 80.12%, which was higher than other experimental conditions.*

[Issue 15] The abbreviation ‘SPD’ on Line 76 appears only once in the entire manuscript. It is recommended not to specify abbreviations for terms that appear infrequently.

[Response] **Accepted.** We have omitted the full meaning of the abbreviation ‘SPD’ in the revised manuscript.

- **Line 75, P.4:** *Saleem et al. illustrated that the SPD plasma-based technology could mineralize 47% of fluorine from PFOA solutions.*

[Issue 16] It is recommended to remove the 3D plot presented in the area introducing the RSM in Figure 1A. The labels on each axis are too small to read. Replacing them with simplified icons representing each task is also recommended.

[Response] **Acknowledged.** As pointed by the reviewer, the labels of RSM plot are relatively small and may be challenging to read. However, this 3D plot primarily serves as a visual representation of the RSM model, and its labels are not pivotal to this main objective. After careful consideration, we have determined that there is no other graphical representation appears to be more suitable or representative for conveying the information we intend to communicate. Hence, we keep it there in the revised manuscript.

[Issue 17] The data and caption provided in the supplementary file are incomplete. The details to be fixed are:

- (1) There are no yellow beads in Figure S2, but cyan beads instead, which are not described in the caption. The caption of Figure S3 also still does not define the cyan beads.

[Response] **Accepted.** We have added the description of cyan beads in the caption of Figure S3 and Figure S4.

- **Line 17, P.3:** *Figure S3. DFT calculations on relative energy and structure changes, and energy barriers for the main reactions in chain initiation and transfer stage under steam atmosphere. The color codes for the atoms are red: C, violet: F, yellow: O, cyan: H.*
- **Line 26, P.5:** *Figure S4. DFT calculations on relative energy and structure changes, and energy barriers for the main reactions in chain termination stage under steam atmosphere. The color codes for the atoms are red: C, violet: F, yellow: O, cyan: H.*

(2) It is believed that the final temperature on the horizontal axis of the data shown in Figures 5B and S4B must exceed 4000 K. The current data is rather lower than the previous temperature, and if it is a typo, it needs to be corrected.

[Response] Acknowledged. We appreciate your observation, and it has come to our attention that our initial explanation regarding the MD simulation's initial conditions was not as clear as intended.

Based on the comparison results of optical emission spectra of plasma jet with the standard spectrum, we determined that the core temperatures of the plasma jet at power levels of 22 kW, 24 kW, 26 kW, 28 kW, and 30 kW are 4500K, 4550K, 4650K, 4700K, and 4750K, respectively. Subsequently, through analysis of the 2D temperature distribution of the plasma jet, we established that the temperatures within the plasma reaction zone correspond to approximately 3300K, 3400K, 3600K, 3800K, and 4000K, corresponding to the respective input powers of 22 kW, 24 kW, 26 kW, 28 kW, and 30 kW.

As a result, we determined the temperature parameters for the reaction systems in the MD simulation as follows: 3300K, 3400K, 3600K, 3800K, and 4000K, corresponding to the respective input powers of 22 kW, 24 kW, 26 kW, 28 kW, and 30 kW. It is worth noting that Figures 5B and S4B presents the stability of the final temperature around 3300K, validating the consistency with the MD simulation's condition settings.

To prevent readers from experiencing the same confusion as the reviewer, the OES measurement and the analysis of two-dimension temperature distribution of plasma torch have been added into the Section 3.1.1 and Fig. 7.

- **Line 506, P.24:** *In order to determine the simulation condition of ReaxFF MD, the core temperature of plasma jet was measured by optical emission spectroscopy (OES) (Optosky ATP2400). The spectra optical emission spectra of plasma jet at different input power ($O_2/C = 0.93$) were shown in Fig. 7. Based on the comparison results of optical emission spectra of plasma jet with the standard spectrum, we determined that the core temperatures of the plasma jet at the input power of 22 kW, 24 kW, 26 kW, 28 kW, and 30 kW are 4500K, 4550K, 4650K, 4700K, and 4750K, respectively. Subsequently, a two-dimensional numerical model based on the DC plasma generator structure (Fig. S11), the core temperature, voltage, and current values of the plasma generator, was developed using Comsol software to simulate the temperature distribution of the plasma jet. As depicted in Figure S12, the gas temperature decreases rapidly as the distance from the center increases from 2mm to 6mm and remains relatively stable within the range of 6 to 10mm. Given that the plasma jet has a diameter of 10mm and the majority of the reaction zones occur at the periphery of plasma jet, we have determined the temperatures in the plasma reaction zone to be approximately 3300K, 3400K, 3600K, 3800K, and 4000K, corresponding to input powers of 22 kW, 24 kW, 26 kW, 28 kW, and 30 kW, respectively. Finally, the activated particle density of plasma jet was investigated by a zero-dimensional numerical modeling. The result of activated particle(s) density of plasma jet was shown in Fig. S13. More detailed measurements methods as well as simulation results on temperature and activated particle density of plasma jet were shown in Text S1, Table S4-S9 and Figure S11-S14.*

Figure 7. The comparison between optical emission spectra of plasma jet and the standard spectrum under different input power (in the wavelengths range of 640nm-760nm): (a) power = 22 kW, O₂/C = 0.9, (b) power = 24 kW, O₂/C = 0.9, (c) power = 26 kW, O₂/C = 0.9, (d) power = 28 kW, O₂/C = 0.9, (e) power = 30 kW, O₂/C = 0.9.

(3) The lower part of the data plot in Figure S8 is cut off, so the horizontal axis and part of the data are not visible.

[Response] Accepted. We have replaced Figure S8 with the complete plot.

(4) Please clearly indicate the information in the sub-figures from (a) to (n) of Figure S10 in the captions.

[Response] Accepted. Thank you for your careful comment. In response, we have included detailed descriptions for sub-figures (a) to (n) of Figure S10 in the revised Supporting Information.

- **Line 948, P.45:** *Figure S11. The comparison between optical emission spectra of plasma jet and the standard spectrum under different input power and atmosphere (in the wavelengths range of 640nm-760nm): (a) power = 22 kW, O₂/C = 0.9, (b) power = 22 kW, H₂O/C = 0.9, (c) power = 22 kW, O₂/C = 2.7, (d) power = 22 kW, H₂O/C = 2.7, (e) power = 22 kW, O₂/C = 4.5, (f) power = 22 kW, H₂O/C = 4.5, (g) power = 24 kW, O₂/C = 0.9, (h) power = 24 kW, H₂O/C = 0.9, (i) power = 26 kW, O₂/C = 0.9, (j) power = 26 kW, H₂O/C = 0.9, (k) power = 28 kW, O₂/C = 0.9, (l) power = 28 kW, H₂O/C = 0.9, (m) power = 30 kW, O₂/C = 0.9, (n) power = 30 kW, H₂O/C = 0.9*

REVIEWERS' COMMENTS

Reviewer #1 (Remarks to the Author):

The authors have answered all my concerns. Thus, I suggest the acceptance of this paper.

Reviewer #2 (Remarks to the Author):

The authors have fully taken into account my comments regarding the accuracy of ReaxFF, and I am supportive of publication in its current form.

Reviewer #3 (Remarks to the Author):

I appreciate the significant effort that the authors have taken to respond to the reviewer queries - and I believe that with these responses and modifications, the manuscript is currently acceptable for publication.

Reviewer #4 (Remarks to the Author):

The authors responded to the best of their ability to technical concerns raised by reviewers. The quality of the manuscript is much improved compared to the previous round. Therefore, the reviewer believes that this manuscript has now reached a publishable standard.

In the following section, we present a detailed itemized response to each comment.

Reviewer # 1

[Issue 1] The authors have answered all my concerns. Thus, I suggest the acceptance of this paper.

[Response] **Thanks a lot. We appreciate your time and effort in assessing our manuscript and are pleased to learn that we have adequately addressed all your concerns.**

Reviewer # 2

[Issue 1] The authors have fully taken into account my comments regarding the accuracy of ReaxFF, and I am supportive of publication in its current form.

[Response] **Thank you very much. We are grateful for your diligent review and positive feedback**

Reviewer # 3

[Issue 1] I appreciate the significant effort that the authors have taken to respond to the reviewer queries - and I believe that with these responses and modifications, the manuscript is currently acceptable for publication..

[Response] **Thank you for your thoughtful evaluation and positive feedback. We sincerely appreciate your recognition of the effort we have invested in addressing the reviewer queries.**

Reviewer # 4

[Issue 1] The authors responded to the best of their ability to technical concerns raised by reviewers. The quality of the manuscript is much improved compared to the previous round. Therefore, the reviewer believes that this manuscript has now reached a publishable standard.

[Response] Thank you for your positive evaluation and recognition of the efforts we put into addressing the technical concerns raised by the reviewers. We are pleased to hear that the quality of the manuscript has significantly improved compared to the previous round.